# Fast and Slow Streams for Online Time Series Forecasting Without Information Leakage

**Ying-yee Ava Lau, Zhiwen Shao, Dit-Yan Yeung**
Department of Computer Science and Engineering
The Hong Kong University of Science and Technology
`yyalau@connect.ust.hk, zhiwen@ust.hk, dyyeung@cse.ust.hk`

## Abstract

Current research in online time series forecasting (OTSF) faces two significant issues. The first is information leakage, where models make predictions and are then evaluated on historical time steps that have already been used in backpropagation for parameter updates. The second is practicality: while forecasting in real-world applications typically emphasizes looking ahead and anticipating future uncertainties, prediction sequences in this setting include only one future step with the remaining being observed time points. This necessitates a redefinition of the OTSF setting, focusing on predicting unknown future steps and evaluating unobserved data points. Following this new setting, challenges arise in leveraging incomplete pairs of ground truth and predictions for backpropagation, as well as in generalizing accurate information without overfitting to noise from recent data streams. To address these challenges, we propose a novel dual-stream framework for online forecasting (DSOF): a slow stream that updates with complete data using experience replay, and a fast stream that adapts to recent data through temporal difference learning. This dual-stream approach updates a teacher-student model learned through a residual learning strategy, generating predictions in a coarse-to-fine manner. Extensive experiments demonstrate its improvement in forecasting performance in changing environments. Our code is publicly available at `https://github.com/yyalau/iclr2025_dsof`.

## 1 Introduction

Accurate forecasts benefit diverse applications like electricity consumption monitoring (Zhu et al., 2024), climate modeling (Mudelsee, 2019), retail (Böse et al., 2017), and stock markets forecasting (Feng et al., 2019). Consequently, significant efforts have focused on better forecasting performance by enhancing deep neural networks to identify complex dependencies in time series data (Zhou et al., 2021; Nie et al., 2023; Wu et al., 2023). An emerging area is online time series forecasting (OTSF) (Anava et al., 2013; Pham et al., 2023), where the forecasting model generates time series sequences using a moving window of 1 and updates the model immediately after corresponding ground truth sequences arrive. This incremental update adapts the model to real-world complications like dynamic datasets and distribution shifts.

### 1.1 Redefining the Online Time Series Forecasting Setting

Existing works that endeavor to improve OTSF model accuracies (Pham et al., 2023; Zhang et al., 2023; 2024a) follow a uniform task setting: at time $t = i$, the model receives input data with timestamps from $t = i - L - H + 2$ to $t = i - H + 1$ and generates predictions from $t = i - H + 2$ to $t = i + 1$; at the next timestamp $t = i + 1$, the actual data for $t = i + 1$ becomes available, allowing the model to evaluate its previous prediction and adjust its parameters based on this new ground truth. For instance, as illustrated in Figure 1, if the input and output windows are of lengths $L = 5$ and $H = 4$ respectively, at time $t = 10$, the model first takes in data with timestamps from $t = 3$ to $t = 7$ and predicts values from $t = 8$ to $t = 11$. Once the actual data for $t = 11$ is available, the model's predictions are evaluated against the ground truth. Subsequently, the ground truth from $t = 8$ to $t = 11$ is used in backpropagation to update the model's parameters. At $t = 12$, the same

procedure performed at $t = 11$ is repeated, advancing predictions by one unit of time. This setting aligns with the conventions of online supervised learning for regression and classification tasks (Hoi et al., 2021): it allows immediate updating of model parameters at the subsequent time step. As the farthest value prediction is one timestamp ahead of the current time, all ground truth data for the predicted sequence is available at the next timestamp.

We raise two concerns about the mentioned setting. First, there is an issue of information leakage. Originally defined in data mining as the use of external data outside the training dataset (Kaufman et al., 2011), information leakage, formally described in Section C, occurs here because the evaluation at $t = 12$ contains data points from $t = 9$ to $t = 11$, which have already been utilized to optimize the model parameters at $t = 11$. This overlap of time steps leads to biased evaluation outcomes, resulting in an overestimation of the model's effectiveness in real-world applications. Second, we question whether it aligns with the general conception of forecasting: forecasting tasks are designed to predict future events (Petropoulos et al., 2022). However, in this scenario, the model only forecasts one future step at any given timestamp. For example, at $t = 10$, it predicts values from $t = 8$ to $t = 11$, even though the ground truth from $t = 8$ to $t = 10$ is already known.

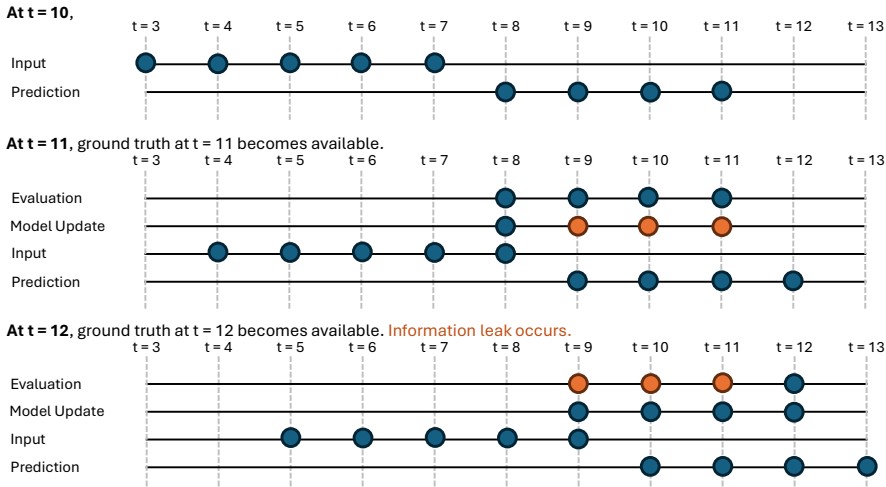

Figure 1: Overview of the data streaming framework in existing online forecasting studies. For illustration, in this figure, we set the lookback window to $L = 5$ and forecast horizon to $H = 4$. Information leak occurs because ground truth from $t = 9$ to $t = 11$ was already used in backpropagation at $t = 11$ and is re-evaluated at $t = 12$.

Consequently, we redefine the OTSF problem to meet two key criteria: the model is evaluated only on time steps without backpropagation for updates, and the output window $H$ should be equal to the number of unknown future time steps to forecast. For example, as illustrated in Figure 2, with input window $L = 5$ and output window $H = 4$, at $t = 10$, the model takes in data from $t = 6$ to $t = 10$ for forecasting values from $t = 11$ to $t = 14$. At $t = 11$, the model can only be updated with ground truths with timestamps up to $t = 11$. The prediction sequence generated at $t = 10$ can only be evaluated on or after $t = 14$, when all actual data from $t = 11$ to $t = 14$ is available.

## 1.2 CHALLENGES AND PROPOSED SOLUTION FOR THE NEW SETTING

A consequent challenge of this new setting is that at $t = 11$, the sequence predicted at $t = 10$ cannot be fully updated in a supervised manner due to the absence of ground truth from $t = 12$ to $t = 14$. This limitation would prevent the model from effectively learning the most up-to-date information in a fully supervised manner, as is typically observed in conventional time series forecasting (Benidis et al., 2022). Moreover, sparse supervisory signals offer limited feedback, hindering the model's ability to learn complex patterns. This can make the model overly sensitive to limited data, which may increase the chance of overfitting if recent data contains noise or anomalies. To address the challenges of the new setting, we further propose an online training framework, called DSOF, that

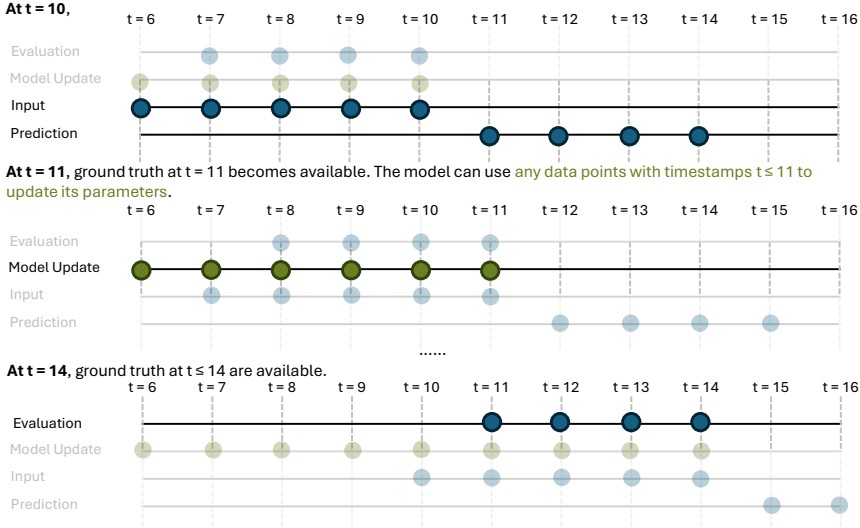

Figure 2: Overview of the redefined OTSF setting. At $t = 10$, the model only forecasts the unknown $H$ future steps from $t = 11$ to $t = 14$. At $t = 11$, it can only update using ground truth observations up to $t = 11$. Predictions made at $t = 10$ can only be evaluated on or after $t = 14$, once all actual data from $t = 11$ to $t = 14$ are available.

leverages the latest and most immediate knowledge from the data stream while maintaining accurate generalization in the data distribution.

Our proposed DSOF framework employs fast and slow streams to update a teacher-student model built on the residual learning approach. Given an input data sequence, the teacher model generates coarse predictions based on historical patterns, while the lightweight student model refines these predictions using real-time feedback. This dual-stream mechanism benefits the online training process in adaptability and stability. The fast stream, inspired by temporal difference (TD) learning's intermediate updates in reinforcement learning, leverages the latest information from data streams to improve near-future forecasts without waiting for the complete ground truth sequence. Meanwhile, the slow stream conducts experience replay (ER) to stabilize training, penalizes overfitting the noise in the incoming data, and ultimately learn accurate generalization of the data distribution. Through extensive experiments, we observe an overall improvement in prediction performance using this integrated strategy across various datasets and forecasting horizons.

## 1.3 CONTRIBUTIONS

In summary, our contributions are as follows:

- We redefine the OTSF setting by restricting the model from predicting and evaluating time steps for which it has already received ground truth feedback. This approach eliminates information leakage and aligns more closely with the forecasting concept as applied in real-world scenarios.

- We develop a novel framework, DSOF, which employs a dual-stream mechanism to integrate real-time data with previously collected data streams for updating model parameters. This approach ensures the maintenance of accurate, current knowledge while also providing robustness and training stability in dynamic environments.

- We conduct extensive experiments to validate that our framework significantly enhances forecasting performance through online updates. Additionally, we provided comprehensive empirical studies that explore various design choices for online training of forecasting models, offering insights into optimizing performance and adaptability in real-time scenarios.

## 2 PRIOR WORKS

**Concept Drift.** Underlying dynamics in real-world time series change over time, leading to differences between training and test data distributions (Tsymbal, 2004). This poses a major challenge for deep neural networks (DNNs) in time series forecasting. To improve DNN robustness against such non-stationarity during test time, various model-agnostic methods and architectural innovations have been introduced. Li et al. (2022) introduced a dynamic data generator for predicting future data distributions, so that models can train on anticipated rather than current distributions. RevIN (Kim et al., 2022) employs reversible instance normalization to adjust for changing statistical properties over time. Building on this, Liu et al. (2023) proposed Slice-level Adaptive Normalization (SAN), which normalizes local temporal slices to handle non-stationarity more granularly. Dish-TS (Fan et al., 2023), using a Dual-CONET framework, can better handle the distributional differences between input and output spaces by separately modeling the distributions of each space. From an architectural perspective, Non-Stationary Transformers, proposed by Liu et al. (2022), combine series stationarization with de-stationary attention to balance predictability and complexity. Additionally, Koopman-based neural operators (Wang et al., 2023; Liu et al., 2024b) have been incorporated to deep learning architectures to better adapt to shifting data distributions. Despite their advancements, however, these methods mainly address evolving trends and fail to learn new patterns that emerge during test time, necessitating research into real-time online learning for non-stationary time series forecasting.

**Online Forecasting.** Unlike traditional forecasting tasks, where training and evaluation are separate phases, online forecasting involves continuous learning over multiple rounds. Anava et al. (2013) pioneered an online forecasting model by updating the parameters of the traditional ARMA model using regret minimization techniques. However, statistical models like ARMA have limited capacity to capture complex temporal dependencies, prompting a shift towards deep learning models for better performance. Pham et al. (2023) proposed FSNet by extending the temporal convolution network (TCN) (Bai et al., 2018) through the introduction of calibration module for weight adjustments and an associative memory for recalling events, enabling quick adaptation to recent changes. As previous models mainly employed only cross-variable approaches, Zhang et al. (2023) proposed OneNet, which integrates cross-variable and cross-time models to capture complex time series dependencies. However, none of the mentioned methods explicitly detect concept drift. The Detect-and-Adapt method (Zhang et al., 2024a) was proposed to first identify concept drift and then adapt the model using a different learning scheme, balancing rapid adjustments with sustained performance. Additionally, research on reducing the computational cost of online updates has emerged, utilizing techniques like hyperdimensional computing (Mejri et al., 2024) and the Moore-Penrose inverse (Zhang et al., 2024b) for quicker online forecasting on lightweight devices. Nonetheless, all of these deep learning approaches failed to address the issue of information leakage when using online learning for time series forecasting tasks, allowing the model to predict and evaluate data points that have already been observed.

## 3 METHODOLOGY

In this section, we introduce our proposed framework, DSOF, which facilitates the online training of time series deep learning models through fast and slow data streams. We describe how this dual-stream mechanism is utilized to update the teacher-student model.

### 3.1 PROBLEM DEFINITION

**Terminology.** Consider a time series dataset $\boldsymbol{X} = (\boldsymbol{x}_1, \cdots, \boldsymbol{x}_{N_{\text{data}}}) \in \mathbb{R}^{N_{\text{data}} \times n}$, comprising $N_{\text{data}}$ observations, each with $n$ dimensions, where $\boldsymbol{x}_i \in \mathbb{R}^n$ represents the data point at time $i$. Given a look-back window of length $L$ that ends at time $i$, denoted as $X_{i-L+1:i} = (\boldsymbol{x}_{i-L+1}, \boldsymbol{x}_{i-L+2}, \cdots, \boldsymbol{x}_i)$, the task is to predict the following $H$ time steps of the series. The model's prediction at time $i$ for the subsequent $H$ steps is represented as $\hat{X}_{i+1:i+H}^{(i)} = (\hat{\boldsymbol{x}}_{i+1}, \hat{\boldsymbol{x}}_{i+2}, \cdots, \hat{\boldsymbol{x}}_{i+H}) = f^{(i)}(X_{i-L+1:i})$.

**Training Phases.** We divide the dataset into two segments, using each segment in different training phases. In the first phase, referred to as the batch learning phase, uses data with timestamps up

to $N_{\text{batch}}$ to train the model in a batch learning manner. In the second phase, referred to as the online learning phase, we use data points with timestamps after $N_{\text{batch}}$ to simulate a real-world scenario of sequential data streaming, updating the model in real time. Generating sequences with a moving window of 1, consequently, the number of online data sequences is given by $N_{\text{online}} = N_{\text{data}} - N_{\text{batch}} - H + 1$.

**Redefined Setting in the Online Phase.** As mentioned in Section 1.1, we redefine the OTSF problem so that the model is evaluated only on time steps without backpropagation for updates, and the output window size should align with the number of unknown future steps to forecast. Formally, with input window $L$ and output window $H$, at $t = i$, the model takes in data from $t = i - L + 1$ to $t = i$ and forecasts values from $t = i + 1$ to $t = i + H$. At $t = i + 1$, the model can only be updated with ground truths with timestamps $t \leq i + 1$. The prediction sequence generated at $t = i$ is evaluated only at $t \geq i + H$, when all actual data from $t = i + 1$ to $t = i + H$ is available.

**Objective and Evaluation Criterion.** Our objective is to minimize the cumulative mean-squared loss, which also serves as the evaluation criterion, expressed as in Equation 1. It is important to note that the model does not evaluate the predicted time steps that have already been observed. Specifically, at time $i$, the prediction $\hat{X}_{i+1:i+H}$ is made without access to the ground truth for any time step greater than $t = i$. The entire prediction sequence is evaluated only on or after $t = i + H$, when all ground truth points are available.

$$\min \frac{1}{N_{\text{online}}} \sum_{i=N_{\text{batch}}}^{N_{\text{data}}-H} \|f^{(i)}(X_{i-L+1:i}) - X_{i+1:i+H}\|_2^2 \tag{1}$$

## 3.2 Two models: Residual Learning Strategy

The framework comprises a teacher-student model that is built on the residual learning strategy. The teacher model, with parameters denoted as $\boldsymbol{\theta}^{(T)}$, usually a robust backbone like DLinear (Zeng et al., 2023) or PatchTST (Nie et al., 2023), generates coarse predictions for given input sequences, as in Equation 2.

$$\hat{X}_{i+1:i+H}^{(T,i)} = f_T^{(i)}(X_{i-L+1:i}; \boldsymbol{\theta}^{(T)}) \tag{2}$$

The student model, with its parameters represented as $\boldsymbol{\theta}^{(S)}$, often a lightweight model such as a Multi-Layer Perceptron (MLP), refines these coarse predictions. By concatenating the lookback sequence and the teacher model's predictions as inputs, as indicated in Equation 3, the student model estimates the error between the teacher's predictions and the ground truth data.

$$\hat{X}_{i+1:i+H}^{(S,i)} = f_S^{(i)}(\text{Concat}(X_{i-L+1:i}, \hat{X}_{i+1:i+H}^{(T,i)}); \boldsymbol{\theta}^{(S)}) \tag{3}$$

The final forecasting prediction is obtained by summing the predictions from both models, as demonstrated in Equation 4.

$$\hat{X}_{i+1:i+H}^{(i)} = f^{(i)}(X_{i-L+1:i}; \boldsymbol{\theta}^{(T)}, \boldsymbol{\theta}^{(S)}) = \hat{X}_{i+1:i+H}^{(T,i)} + \hat{X}_{i+1:i+H}^{(S,i)} \tag{4}$$

## 3.3 Two Data Streams: Fast and Slow

This section presents the two data streams for updating the teacher-student model in Section 3.2: a slow stream using complete ground truth via ER and a fast stream incorporating latest data through a TD learning-inspired method. The term "TD" indicates that our approach is inspired by TD learning in reinforcement learning, particularly in estimating the ground truth values at the next time step, addressing the challenge of missing ground truth in the redefined OTSF setting.

### 3.3.1 Slow Data Stream with Experience Replay

ER is a common approach to address catastrophic forgetting in continual learning. Unlike continual learning which aims for effective performance on both previously learned and newly acquired tasks (Rolnick et al., 2019), and reinforcement learning which seeks to break correlations from samples taken at adjacent trajectories to enhance network convergence (Mnih et al., 2015), our focus is on adapting to the evolving temporal domain, aligning with the objective of minimizing cumulative

mean squared error (MSE). Thus, instead of promoting diversity in sample patterns, the replay buffer $\mathcal{B}$ retains only the most recent samples.

The replay buffer has a fixed size $N_\mathcal{B}$, and operates on a first-in, first-out basis. At time $t = i$, upon receiving the ground truth data $\boldsymbol{x}_i$, the buffer appends the pairing of the input sequence $X_{i-H-L-1:i-H}$ and output sequence $X_{i-H+1:i}$. If the buffer exceeds $N_\mathcal{B}$, the oldest sample, with the input sequence $X_{i-N_\mathcal{B}-H-L+1:i-N_\mathcal{B}-H}$ and output sequence $X_{i-N_\mathcal{B}-H+1:i-N_\mathcal{B}}$, is removed. This process is referred to as the slow update stream, as it waits for all ground truth values of the output horizon to arrive before adding the data stream to the buffer. This setup resembles a time-delayed environment in reinforcement learning (Liotet et al., 2022; Wu et al., 2024), where new predictions must be made despite delayed feedback from the environment.

### 3.3.2 Fast Data Stream with Temporal Difference Loss

Performing ER in Section 3.3.1 requires complete ground truth data for all time steps in the data stream. A major drawback is that with a large prediction length $H$, we must wait for $H$ time steps for all ground truth values to arrive, before adding the sequence to the replay buffer $\mathcal{B}$. However, in the presence of concept drift, we aim to update the model with immediate observations — even without complete ground truth for the entire forecasting horizon — to improve near-future predictions.

At time $i - 1$, the teacher-student model generates $\hat{X}_{i:i+H-1}^{(i-1)}$. Right after the prediction, at time $i$, the ground truth $\boldsymbol{x}_i$ becomes available. To compensate for the absence of ground truth for the future timestamps $t = i + 1$ to $t = i + H - 1$, the teacher model's prediction at time $t = i$, i.e. $\hat{X}_{i+1:i+H}^{(T,i)}$, is utilized to generate the pseudo labels $\tilde{X}_{i:i+H-1}^{(i)}$, as outlined in Equation 5.

$$\tilde{X}_{i:i+H-1}^{(i)} = [\boldsymbol{x}_i, \hat{X}_{i+1,i+H-1}^{(T,i)}] \tag{5}$$

In practice, the pseudo labels can be generated with different proportions of ground truth data and teacher model predictions. Consequently, we extend Equation 5 by introducing a parameter $k$, which signifies the use of the latest $k$ ground truth data points along with the initial $H - k$ steps of predictions from the teacher model to form the pseudo label. The impact of using different $k$ values, as well as utilizing multiple $k$ values, are discussed in Section A.6.2.

It is assumed that the teacher model's predictions for the distant future are considerably more uncertain and less significant to the current prediction than those for the near future. Considering the temporal nature of time series data, we introduce a geometric decay factor $\gamma$, to reduce the influence of prediction errors and pseudo labels for each step further from the current observation, as outlined in Equation 6. We further elaborate on establishing the relationship between the TD approach used in our method and that found in reinforcement learning in Section B.

$$\ell_{TD}^{(i-1)}(\hat{X}_{i:i+H-1}^{(i-1)}, \tilde{X}_{i:i+H-1}^{(i)}) = \frac{1}{H} \sum_{j=1}^{H} \gamma^{j-1} \|\hat{\boldsymbol{x}}_j^{(i)} - \tilde{\boldsymbol{x}}_j^{(i)}\|^2 \tag{6}$$

### 3.4 Overview of the Framework in Online Training

This section outlines our framework in the online training stage, with a detailed algorithm presented in Algorithm 1. During the online training phase, upon receiving each new data point, we first update the replay buffer $\mathcal{B}$ and then conduct an ER, as detailed in Section 3.3.1. This process involves randomly selecting a small batch of size $N_b$ from the buffer to train and update the teacher-student model. By default, ER is performed whenever a new data point arrives, but this frequency can be adjusted — for instance, it can be triggered twice per new data point or once for every two data points. After completing ER, the student model is updated using the TD loss described in Section 3.3.2. To maintain training stability, the teacher model's parameters $\boldsymbol{\theta}^{(T)}$ remain fixed, while the student model is updated with pseudo labels.

---

**Algorithm 1** Overview of DSOF during the online training phase.

---

1: **Initialize:**
   Replay Buffer $\mathcal{B} \leftarrow \varnothing$ with capacity $N_{\mathcal{B}}$
   Batch Size $N_b$
   Learning rates $\alpha_T, \alpha_S, \alpha_O$
2: **for** $i = N_{\text{batch}} + 1$ to $N_{\text{data}} - H$ **do**
3:     Received ground truth $x_i$.
4:     $\mathcal{B} \leftarrow \mathcal{B} \cup \{(X_{i-H-L+1:i-H}, X_{i-H+1:i})\}$           ▷ *Append the newest sequence.*
5:     **if** $|\mathcal{B}| \geq N_{\mathcal{B}}$ **then**           ▷ *The buffer exceeds its capacity.*
6:        $\mathcal{B}: \mathcal{B} \leftarrow \mathcal{B} \setminus \{(X_{i-N_{\mathcal{B}}-H-L+1:i-N_{\mathcal{B}}-H}, X_{i-N_{\mathcal{B}}-H+1:i-N_{\mathcal{B}}})\}$    ▷ *Remove the oldest sequence.*
7:     **end if**
8:     **if** $|\mathcal{B}| \geq N_b$ **then**           ▷ *The buffer has enough sequences to sample.*
9:        $\boldsymbol{\theta}^{(T)}, \boldsymbol{\theta}^{(S)} \leftarrow \text{EXPERIENCE\_REPLAY\_UPDATE}(\mathcal{B}, \boldsymbol{\theta}^{(T)}, \boldsymbol{\theta}^{(S)})$
10:    **end if**
11:     $\boldsymbol{\theta}^{(S)} \leftarrow \text{TEMPORAL\_DIFFERENCE\_UPDATE}(X_{i-L:i}, \hat{X}_{i:i+H-1}^{(i-1)}, \boldsymbol{\theta}^{(T)}, \boldsymbol{\theta}^{(S)})$
12:     $\hat{X}_{i+1:i+H}^{(i)} = f^{(i)}(X_{i-L+1:i}; \boldsymbol{\theta}^{(T)}, \boldsymbol{\theta}^{(S)})$       ▷ *Teacher-student model forecasts the next step.*
13: **end for**

14: **procedure** EXPERIENCE_REPLAY_UPDATE$(\mathcal{B}, \boldsymbol{\theta}^{(T)}, \boldsymbol{\theta}^{(S)})$
15:     $(\mathcal{X}_L, \mathcal{X}_H) \sim \mathcal{B}$           ▷ *Sample a mini-batch from the buffer.*
16:     $\hat{\mathcal{X}}_H = f^{(i-1)}(\mathcal{X}_L; \boldsymbol{\theta}^{(T)}, \boldsymbol{\theta}^{(S)})$ (Equation 4)     ▷ *Teacher-student model makes a prediction.*
17:     $\boldsymbol{\theta}^{(T)} \leftarrow \boldsymbol{\theta}^{(T)} - \alpha_T \nabla_{\boldsymbol{\theta}^{(T)}} \ell(\hat{\mathcal{X}}_H, \mathcal{X}_H)$       ▷ *Update teacher model's parameters.*
18:     $\boldsymbol{\theta}^{(S)} \leftarrow \boldsymbol{\theta}^{(S)} - \alpha_S \nabla_{\boldsymbol{\theta}^{(S)}} \ell(\hat{\mathcal{X}}_H, \mathcal{X}_H)$       ▷ *Update student model's parameters.*
19:     **return** $\boldsymbol{\theta}^{(T)}, \boldsymbol{\theta}^{(S)}$
20: **end procedure**

21: **procedure** TEMPORAL_DIFFERENCE_UPDATE$(X_{i-L+1:i}, \hat{X}_{i:i+H-1}^{(i-1)}, \boldsymbol{\theta}^{(T)}, \boldsymbol{\theta}^{(S)})$
22:     Freeze the teacher parameters $\boldsymbol{\theta}^{(T)}$.
23:     $\hat{X}_{i+1:i+H}^{(T,i)} = f_T^{(i)}(X_{i-L+1:i}; \boldsymbol{\theta}^{(T)})$ (Equation 2)    ▷ *Teacher model makes a prediction utilizing $x_i$.*
24:     $\tilde{X}_{i:i+H-1}^{(i)} = [x_i, \hat{X}_{i+1:i+H-1}^{(T,i)}]$ (Equation 5)        ▷ *Generate the pseudo-label.*
25:     Compute the temporal difference loss $\ell_{TD}^{(i-1)}(\hat{X}_{i:i+H-1}^{(i-1)}, \tilde{X}_{i:i+H-1}^{(i)})$ using Equation 6.
26:     $\boldsymbol{\theta}^{(S)} \leftarrow \boldsymbol{\theta}^{(S)} - \alpha_O \nabla_{\boldsymbol{\theta}^{(S)}} \ell_{TD}^{(i-1)}(\hat{X}_{i:i+H-1}^{(i-1)}, \tilde{X}_{i:i+H-1}^{(i)})$    ▷ *Update student model's parameters.*
27:     **return** $\boldsymbol{\theta}^{(S)}$
28: **end procedure**

---

## 4 EXPERIMENTS

In this section, we demonstrate the effectiveness and adaptability of our proposed method. Our experiments involve comparing our framework against batch learning approaches, utilizing a diverse range of model architectures as baselines. We also evaluate the effectiveness of our method against other online learning frameworks, which we have modified to avoid information leakage. Further analyses are conducted in Section A.

### 4.1 GENERAL FORECASTING RESULTS

In this section, DSOF is compared to batch learning, demonstrating the advantages of online training and the framework's ability to adapt batch methods to online scenarios for improved performance.

**Datasets.** We utilize benchmarks from Autoformer (Wu et al., 2021), which are primarily designed for long-term forecasting tasks. For our experiments, we adhere to the data distribution outlined in Pham et al. (2023), splitting the dataset chronologically into 20% for training, 5% for validation, and 75% for testing, to simulate the temporal distribution shift in real world scenarios. The datasets used in our experiments include the following: the **Electricity (ECL)** dataset tracks the electricity consumption of 321 clients from 2012 to 2014; the **ETT** benchmark tracks oil temperature and six power load features over two years, with **ETTh2** recorded hourly and **ETTm1** at 15-minute intervals; the **Exchange (Ex.)** dataset includes daily exchange rates for eight countries from 1990 to

2016; the **Weather** dataset, collected every 10 minutes throughout 2020, features 21 meteorological indicators, such as air temperature and humidity; the **Traffic** dataset provides hourly road occupancy rates from the California Department of Transportation, reflecting measurements from various sensors on freeways in the San Francisco Bay Area. Detailed information about the datasets, including the number of features and timesteps, is presented in Table 1.

Table 1: Statistics of benchmark datasets used.

|  | Electricity | ETTh2 | ETTm1 | Exchange | Traffic | Weather |
|---|---|---|---|---|---|---|
| Features | 321 | 7 | 7 | 7 | 862 | 21 |
| Timesteps | 26304 | 14400 | 57600 | 7396 | 17544 | 52696 |

**Baselines.** We set prediction lengths $H$ to 1, 24, and 48, with a lookback length of 96, following the experimental setting used in the previous works (Pham et al., 2023; Zhang et al., 2023). In our results, batch learning depends exclusively on the teacher model, whereas our method integrates a three-layered MLP with a hidden dimension of 16 as the student model, built on top of the teacher model. A variety of backbone architectures for the teacher model are utilized for comparison with batch learning. These included linear backbones such as **DLinear** (Zeng et al., 2023), **FITS** (Xu et al., 2024), and **TimeMixer** (Wang et al., 2024). Additionally, we employed convolution backbones including **FSNet** (Pham et al., 2023) and **OneNet** (Zhang et al., 2023). Lastly, for transformer backbones, we used **iTransformer (iTrans.)** (Liu et al., 2024a), **PatchTST** (Nie et al., 2023), and **Non-Stationary Transformer (NSTrans)** (Liu et al., 2022).

**Results.** To account for variability from random weight initialization, we ran experiments with five random seeds. Table 2 reports the average MSE, while standard deviations are in Section E.1. Our framework generally outperforms batch learning across datasets and architectures, effectively addressing distribution shifts in online forecasting. Notably, on the ECL dataset, FSNet and NSTransformer as teacher models achieve over a 50% MSE reduction. However, in some cases, batch learning outperforms online approaches. One possible factor affecting performance could be the normalization choices and hyperparameters of the backbone, which are kept the same as in batch learning and not further optimized for online learning.

**Visualizations.** Figure 3 presents visualizations of forecasting results, illustrating a 48-step prediction sequence on the Traffic dataset. It is observed that at step 300, where the distribution closely resembles the training distribution, the performance of batch learning is similar to that of online learning. However, at later time steps, such as 5000 and beyond, the performance of online learning surpasses that of batch learning.

## 4.2 EVALUATING OTSF FRAMEWORKS WITHOUT INFORMATION LEAKAGE

**Baselines.** Previous works, such as FSNet (Pham et al., 2023) and OneNet (Zhang et al., 2023), adopted a different OTSF setting that suffers from information leakage, leading to an overestimation of model effectiveness during evaluation. To ensure a fair comparison, we modified this setting to eliminate information leakage by delaying ground truth feedback. At time $t = i$, the model receives data from $t = i - L + 1$ to $t = i$, and generates prediction with time steps from $t = i + 1$ to $t = i + H$. Unlike before, where predictions were evaluated and immediately backpropagated using the ground truth at $t = i + 1$, this procedure is performed after $H$ steps, specifically at $t = i + H$. Here, we name this framework as "Delayed Gradient" (DGrad).

We also incorporated continual learning methods, TFCL (Aljundi et al., 2019) and DER++ (Buzzega et al., 2020), for comparison. However, these approaches are not tailored for time series forecasting. They were originally utilized in vision tasks rather than forecasting and do not account for labels with temporal contexts. Consequently, the techniques they propose may also be susceptible to information leakage. We mitigate this risk by delaying the ground truth feedback, employing the same strategy as in DGrad.

While DSOF includes a student model, the methods, DGrad, TFCL, and DER++ do not. To ensure a fair comparison, we additionally present an alternative version of DSOF that excludes the student model. In this version, instead of combining the outputs from both the teacher and student models, we rely solely on the teacher model's output — this modifies the prediction results from Equation 4

Table 2: Comparison of MSE results between our framework and batch learning across various datasets and backbones. Our method integrates a MLP as the student model built on top of the teacher model, while batch learning relies exclusively on the teacher model. The better results from the two settings are highlighted in bold.

| | H | DLinear Batch Learn. | DLinear DSOF | FITS Batch Learn. | FITS DSOF | FSNet Batch Learn. | FSNet DSOF | OneNet Batch Learn. | OneNet DSOF | iTrans. Batch Learn. | iTrans. DSOF | PatchTST Batch Learn. | PatchTST DSOF | NSTrans. Batch Learn. | NSTrans. DSOF |
|---|---|---|---|---|---|---|---|---|---|---|---|---|---|---|---|
| ECL | 1 | 2.842 | **2.065** | 2.870 | **2.235** | 3.8e+2 | **2.330** | 3.2e+1 | **4.733** | **1.976** | 2.430 | 4.770 | **2.244** | 3.9e+1 | **2.703** |
| ECL | 24 | 1.5e+1 | **4.737** | **4.509** | 4.597 | 4.7e+2 | **5.475** | 8.5e+1 | **4.510** | **4.119** | 5.155 | 1.7e+1 | **5.169** | 4.0e+1 | **8.668** |
| ECL | 48 | 2.7e+1 | **6.181** | **5.257** | 5.433 | 4.8e+2 | **7.000** | 1.5e+2 | **5.943** | **4.936** | 6.015 | 1.7e+1 | **6.665** | 3.8e+1 | **9.981** |
| ETTh2 | 1 | 0.470 | **0.365** | 0.522 | **0.375** | 1.1e+1 | **0.431** | 2.531 | **0.548** | 0.872 | **0.384** | 0.915 | **0.382** | 3.088 | **0.415** |
| ETTh2 | 24 | 2.269 | **1.701** | 2.189 | **1.757** | 1.9e+1 | **3.114** | 7.017 | **2.363** | 2.688 | **1.869** | 5.213 | **1.925** | 3.973 | **2.127** |
| ETTh2 | 48 | 3.389 | **3.082** | 3.275 | **2.988** | 2.3e+1 | **5.318** | 9.790 | **4.037** | 3.769 | **3.465** | 6.566 | **4.473** | 4.840 | **4.729** |
| ETTm1 | 1 | 0.112 | **0.105** | 0.123 | **0.111** | 0.190 | **0.121** | 0.156 | **0.096** | 0.179 | **0.152** | 0.155 | **0.108** | 1.101 | **0.141** |
| ETTm1 | 24 | 0.628 | **0.525** | 0.732 | **0.542** | 1.500 | **0.609** | 1.094 | **0.418** | 1.049 | **0.618** | 1.170 | **0.582** | 1.503 | **0.904** |
| ETTm1 | 48 | 0.818 | **0.695** | 0.900 | **0.716** | 2.279 | **0.843** | 1.588 | **0.554** | 1.320 | **0.856** | 2.103 | **0.805** | 1.547 | **1.119** |
| Ex. | 1 | **0.009** | **0.009** | 0.012 | **0.011** | 0.034 | **0.010** | 0.024 | **0.009** | 0.011 | **0.010** | 0.024 | **0.011** | 0.234 | **0.015** |
| Ex. | 24 | 0.098 | **0.095** | 0.095 | **0.093** | 0.754 | **0.120** | 0.487 | **0.152** | 0.117 | **0.110** | 0.443 | **0.103** | 0.335 | **0.127** |
| Ex. | 48 | 0.194 | **0.192** | 0.178 | **0.176** | 1.366 | **0.270** | 0.815 | **0.289** | **0.209** | 0.218 | 1.107 | **0.213** | 0.449 | **0.280** |
| Traffic | 1 | **0.302** | **0.302** | 0.342 | **0.313** | 0.599 | **0.228** | 0.265 | **0.264** | 0.243 | **0.242** | 0.280 | **0.232** | 0.762 | **0.257** |
| Traffic | 24 | 0.649 | **0.608** | 0.617 | **0.607** | 0.761 | **0.366** | 0.576 | **0.327** | 0.459 | **0.422** | 0.573 | **0.432** | 1.102 | **0.565** |
| Traffic | 48 | 0.769 | **0.681** | 0.707 | **0.680** | 0.824 | **0.403** | 0.683 | **0.369** | 0.517 | **0.457** | 0.621 | **0.454** | 1.001 | **0.604** |
| Weather | 1 | 0.357 | **0.337** | 0.359 | **0.341** | 0.728 | **0.388** | 0.478 | **0.296** | 0.474 | **0.336** | 0.423 | **0.361** | 2.489 | **0.336** |
| Weather | 24 | 1.182 | **1.043** | 1.236 | **1.086** | 1.760 | **1.020** | 1.377 | **0.671** | 1.492 | **1.019** | 1.494 | **0.877** | 2.881 | **1.037** |
| Weather | 48 | 1.702 | **1.447** | 1.634 | **1.434** | 2.620 | **1.415** | 2.699 | **0.909** | 1.740 | **1.435** | 2.033 | **1.308** | 3.039 | **1.419** |

to Equation 7, while keeping other parts of the algorithm unchanged. We denote this version as "DSOF (w/o $\boldsymbol{\theta}^{(S)}$)".

$$\hat{X}^{(i)}_{i+1:i+H} = \hat{X}^{(T,i)}_{i+1:i+H} = f^{(i)}_T(X_{i-L+1:i}; \boldsymbol{\theta}^{(T)}) \tag{7}$$

**Results.** Similar to the comparison with batch learning in Section 4.1, experiments were repeatedly trained using five different random seeds. Table 3 shows the results, where DSOF, with or without the student model, consistently ranks among the top two methods. Notably, for ECL at a prediction length of 48, our approach reduces MSE by at least 15%, regardless of the teacher model used, outperforming both the baseline and typical continual learning methods.

In some cases, omitting the student model yields better results. While its inclusion does not always guarantee optimal performance, our study in Section A.7 suggests it is less sensitive to learning rate variations, reducing hyperparameter tuning efforts.

## 5 CONCLUSION

In summary, we first redefined the online forecasting setting to eliminate information leakage, ensuring that the model does not predict or evaluate time steps for which it has already received ground truth feedback. We then introduced a model-agnostic framework, called DSOF, enabling various types of models to adapt to temporal shifts in real-time data. Our extensive experiments demonstrate that DSOF significantly improves forecasting performance through ongoing updates. Particularly, when compared to batch learning, the ECL dataset demonstrates the most significant improvement, with teacher model backbones such as FSNet and NSTransformer, achieving over a 50% reduction in MSE. Additionally, we have provided detailed ablation studies on design choices for training forecasting models online, offering valuable insights for better forecasting performance and adaptability. However, there are still areas to explore, such as identifying better model architectures suited for online forecasting, using more advanced semi-supervised techniques to generate pseudo labels, improving ER methods, and extending forecasting to longer time periods.

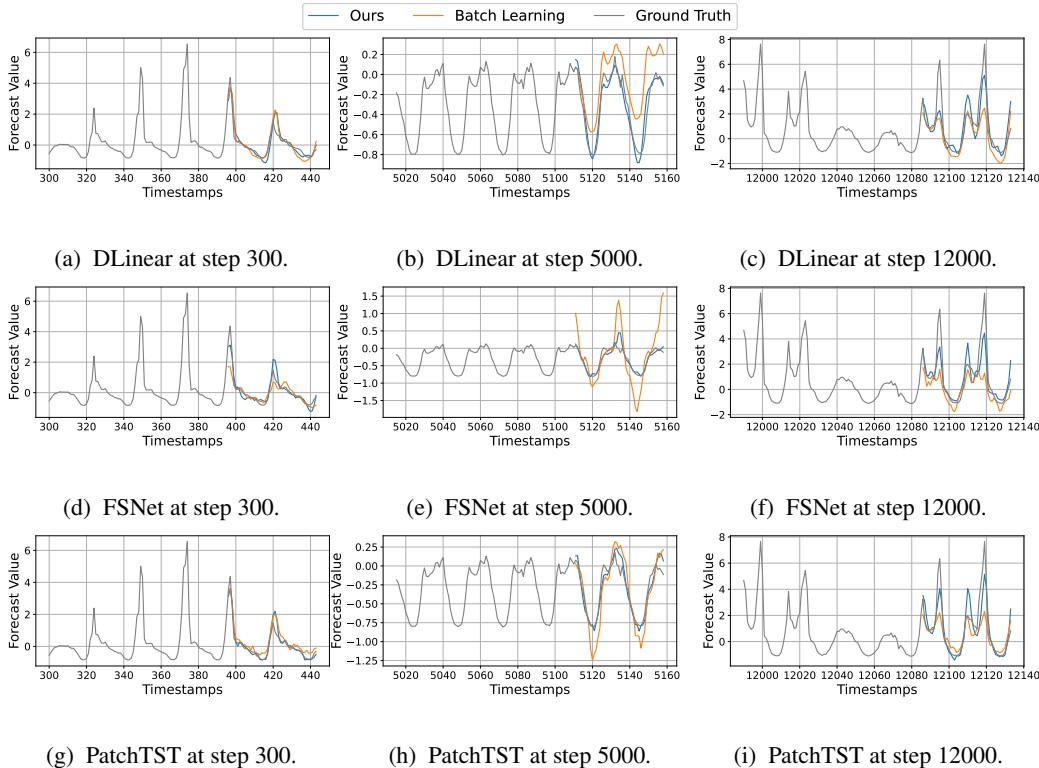

(a) DLinear at step 300.      (b) DLinear at step 5000.      (c) DLinear at step 12000.

(d) FSNet at step 300.      (e) FSNet at step 5000.      (f) FSNet at step 12000.

(g) PatchTST at step 300.      (h) PatchTST at step 5000.      (i) PatchTST at step 12000.

Figure 3: Visualization of the online learning process for Traffic at steps 300, 5000 and 12000.

Table 3: Comparison of the MSE results between DGrad, i.e. the modified online training frameworks from earlier studies (Pham et al., 2023; Zhang et al., 2023), as well as TFCL (Aljundi et al., 2019) and DER++ (Buzzega et al., 2020). All configurations have been modified to prevent information leakage.

| Teacher Model | Student Model | Framework | ECL 1 | 24 | 48 | ETTh2 1 | 24 | 48 | Traffic 1 | 24 | 48 |
|---|---|---|---|---|---|---|---|---|---|---|---|
| DLinear | ✗ | DGrad | 2.187 | 9.954 | 1e+01 | 0.386 | 3.038 | 6.251 | 1e+15 | 1.189 | 1.461 |
| | ✗ | TFCL | 3.033 | 6.897 | 9.746 | 0.718 | 2.240 | 3.457 | 0.324 | 0.697 | 0.745 |
| | ✗ | DER++ | 2.172 | 5.369 | 7.339 | 0.385 | 1.986 | 3.168 | 0.300 | 0.623 | 0.709 |
| | ✗ | DSOF (w/o $\theta^{(S)}$) | 2.066 | 4.759 | 6.250 | **0.365** | **1.694** | **3.015** | **0.299** | **0.603** | 0.685 |
| | MLP | DSOF | **2.065** | **4.737** | **6.181** | **0.365** | 1.701 | 3.082 | 0.302 | 0.608 | **0.681** |
| FSNet | ✗ | DGrad | 3.197 | 5e+01 | 5e+01 | 0.470 | 2.956 | 5.359 | 1e+18 | 0.470 | 0.574 |
| | ✗ | TFCL | 2.968 | 7.482 | 8.512 | 0.726 | 3.469 | 6.256 | 0.315 | 0.418 | 0.467 |
| | ✗ | DER++ | 2.902 | 1e+01 | 1e+01 | 0.432 | **2.726** | 5.239 | 0.286 | 0.411 | 0.449 |
| | ✗ | DSOF (w/o $\theta^{(S)}$) | 2.567 | 5.611 | 7.130 | **0.427** | 2.944 | **4.846** | 0.265 | **0.364** | **0.394** |
| | MLP | DSOF | **2.330** | **5.475** | **7.000** | 0.431 | 3.114 | 5.318 | **0.228** | 0.366 | 0.403 |
| PatchTST | ✗ | DGrad | 3.863 | 9.453 | 8.607 | 0.392 | 2.759 | 5.721 | 0.251 | 0.446 | 0.480 |
| | ✗ | TFCL | 3.805 | 7.653 | 1e+01 | 0.714 | 3.275 | 7.763 | 0.272 | 0.558 | 0.607 |
| | ✗ | DER++ | 3.403 | 6.846 | 8.025 | 0.386 | 2.163 | 5.849 | 0.242 | 0.478 | 0.502 |
| | ✗ | DSOF (w/o $\theta^{(S)}$) | 3e+03 | 5.253 | **6.427** | **0.373** | **1.838** | **3.568** | **0.231** | 0.434 | 0.459 |
| | MLP | DSOF | **2.244** | **5.169** | 6.665 | 0.382 | 1.925 | 4.473 | 0.232 | **0.432** | **0.454** |

# 6 ACKNOWLEDGEMENTS

This work has been made possible by a Research Impact Fund project (R6003-21) and an Innovation and Technology Fund project (ITS/004/21FP) funded by the Hong Kong Government.

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

Table 4: Wall time comparison across various frameworks and datasets, measured in milliseconds per iteration (ms/itr). Lower values indicate higher efficiency.

| | ECL | | ETTh2 | | Traffic | |
|---|---|---|---|---|---|---|
| Framework | Mean | SD | Mean | SD | Mean | SD |
| DGrad | 5.58 | 0.18 | 4.63 | 0.14 | 11.03 | 1.06 |
| DER++ | 13.79 | 1.08 | 12.71 | 3.50 | 23.82 | 3.37 |
| TFCL | 23.28 | 1.69 | 13.02 | 0.33 | 30.89 | 2.76 |
| DSOF (w/o $\boldsymbol{\theta}^{(S)}$) | 23.16 | 1.40 | 8.13 | 0.12 | 51.66 | 29.09 |
| DSOF | 30.76 | 1.06 | 12.04 | 0.16 | 63.96 | 12.42 |

Table 5: Processor time comparison across various frameworks and datasets, measured in ms/itr. Lower values indicate higher efficiency.

| | ECL | | ETTh2 | | Traffic | |
|---|---|---|---|---|---|---|
| Framework | Mean | SD | Mean | SD | Mean | SD |
| DGrad | 4.90 | 0.36 | 3.67 | 0.18 | 166.87 | 19.09 |
| DER++ | 193.29 | 17.46 | 155.89 | 26.68 | 373.74 | 30.84 |
| TFCL | 337.87 | 17.38 | 137.98 | 3.51 | 551.64 | 79.99 |
| DSOF (w/o $\boldsymbol{\theta}^{(S)}$) | 358.31 | 18.71 | 7.32 | 0.32 | 596.36 | 58.51 |
| DSOF | 478.45 | 31.96 | 10.91 | 1.20 | 800.70 | 10.74 |

## A MODEL DESIGN CHOICES AND ANALYSIS

This section examines the effects of different learning strategies on the teacher-student model. It also analyzes experience replay (ER) parameters, including buffer size and update frequency, and it examines the impact of temporal difference (TD) loss on mean squared error (MSE) results. Additionally, it also compares how the framework performs with and without the student model, specifically by examining the influence of the learning rate.

### A.1 COMPUTATIONAL COST COMPARISON OF ONLINE TRAINING FRAMEWORKS

#### A.1.1 RUNTIME COMPARISON

Online learning consists of two phases: training with batches of data and testing with streaming data. We focus on the testing phase, where the model updates parameters each time a new sample arrives. We measure the time each framework takes to perform an online update per sample using an Intel Xeon Silver 4214R processor (12 cores, 24 threads, 2.40 GHz) and an NVIDIA GeForce RTX 3090 GPU (24GB memory).

In this experiment, the teacher model used in DSOF is DLinear. The mean and standard deviation of the runtime are calculated over 250 runs. Wall time and processor time are reported in Table 4 and Table 5, respectively, in milliseconds per iterations (ms/itr). For instance, 100 ms/itr indicates that the framework can perform parameter updates for one sample in 100 ms. Therefore, a lower number indicates better performance. Both tables show that our method, DSOF, is less efficient compared to others, highlighting a weakness of our approach. This inefficiency is especially evident in the Traffic dataset, which, as stated in Table 1, has a significantly larger feature dimensionality than ECL and ETTh2.

Table 6: Comparison of GPU memory usage during the online phase across different online learning frameworks (measured in MiB).

| Framework | ECL | ETTh2 | Traffic |
|---|---|---|---|
| DGrad | 386 | 346 | 436 |
| DER++ | 424 | 346 | 542 |
| TFCL | 392 | 346 | 440 |
| DSOF (w/o $\theta^{(S)}$) | 386 | 346 | 436 |
| DSOF | 400 | 348 | 508 |

### A.1.2 GPU MEMORY USAGE COMPARISON

One downside of our approach compared to other online learning frameworks is that it involves using a student model for residual learning, which adds extra model parameters. As in the other experiments conducted in this study, the student model is a 3-layered MLP with a hidden size of 16, and the replay batch size in the ER procedure is set to 32. Using DLinear as the teacher model for DSOF, the GPU memory is evaluated during the online phase, as shown in Table 6, with units measured in mebibytes (MiB). The results suggest that our method does not require significantly more GPU memory.

### A.2 COMPONENTS OF THE FRAMEWORK: EXPERIENCE REPLAY AND TEMPORAL DIFFERENCE LOSS

#### A.2.1 QUANTITATIVE RESULTS

Experiments are conducted to investigate the necessity of ER and TD loss. Table 7 summarizes the comparison between the MSE performances with and without ER and TD loss. The first row of the table, which represents the setting without either ER or TD loss, corresponds to the method, DGrad, mentioned in Section 4.2. These results verify that both components are critical for the effectiveness of our approach.

Table 7: Impact of ER and TD loss components on model performance.

| Teacher Model | Student Model | ER | TD | ECL 1 | ECL 24 | ECL 48 | ETTh2 1 | ETTh2 24 | ETTh2 48 | Traffic 1 | Traffic 24 | Traffic 48 |
|---|---|---|---|---|---|---|---|---|---|---|---|---|
| DLinear | MLP | ✗ | ✗ | 2.842 | 14.705 | 26.795 | 0.470 | 2.269 | 3.389 | 0.302 | 0.649 | 0.769 |
| | | ✗ | ✓ | 2.118 | 13.530 | 30.725 | 0.523 | 2.199 | 3.535 | **0.300** | 0.650 | 0.762 |
| | | ✓ | ✗ | 2.068 | 5.038 | 6.405 | 0.372 | 1.748 | 3.255 | 0.302 | **0.605** | 0.682 |
| | | ✓ | ✓ | **2.065** | **4.737** | **6.181** | **0.365** | **1.701** | **3.082** | 0.302 | 0.608 | **0.681** |
| FSNet | MLP | ✗ | ✗ | 381.629 | 473.965 | 481.403 | 10.555 | 19.326 | 23.412 | 0.599 | 0.761 | 0.824 |
| | | ✗ | ✓ | 33.746 | 282.664 | 314.527 | 2.296 | 13.387 | 11.011 | 0.400 | 0.667 | 0.764 |
| | | ✓ | ✗ | **2.228** | 5.514 | 7.132 | 0.465 | 3.220 | **5.267** | 0.231 | **0.364** | **0.402** |
| | | ✓ | ✓ | 2.330 | **5.475** | **7.000** | **0.431** | **3.114** | 5.318 | **0.228** | 0.366 | 0.403 |
| PatchTST | MLP | ✗ | ✗ | 4.770 | 16.594 | 16.943 | 0.915 | 5.213 | 6.566 | 0.280 | 0.573 | 0.621 |
| | | ✗ | ✓ | 3.952 | 24.128 | 13.939 | 0.900 | 4.262 | 4.626 | 0.270 | 0.584 | 0.619 |
| | | ✓ | ✗ | 2.409 | **5.109** | 6.978 | **0.378** | 2.022 | 4.514 | **0.232** | **0.432** | 0.456 |
| | | ✓ | ✓ | **2.244** | 5.169 | **6.665** | 0.382 | **1.925** | **4.473** | **0.232** | **0.432** | **0.454** |

#### A.2.2 MSE ANALYSIS

In the ETTh2 dataset, as shown in Figure 4a, DGrad struggles to quickly adapt to new distributions from timestamps 3300 to 3450, resulting in higher errors compared to other methods. The advantages of ER and TD loss suit different situations: between timestamps from 3300 and 3450, using

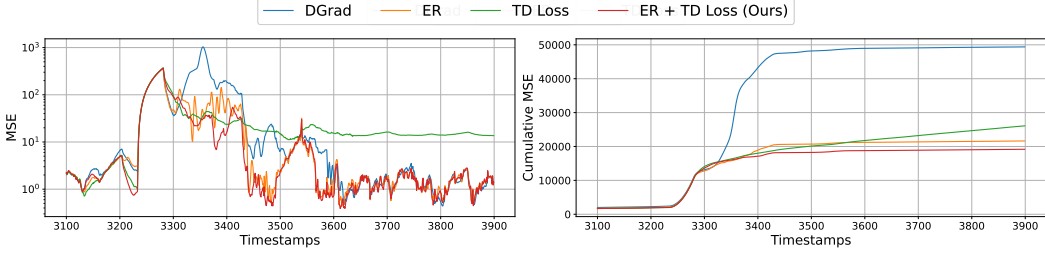

(a) MSE at individual timestamps.

(b) Cumulative squared error over time.

Figure 4: MSE analysis of ER and TD loss components using the ETTh2 dataset.

the TD loss outperforms, whereas ER excels in timestamps from 3450 to 3900. In the long run, ER helps the model generalize with complete labels and batches of data. Again, the cumulative squared error plot in Figure 4b shows that using both components results in slower MSE growth.

### A.2.3 INTERACTION OF THE COMPONENTS

To explore the interaction between ER and TD loss components, we conduct two case studies using the ETTh2 dataset. Experiments are conducted across five runs with different random seeds. In Figure 5, the shaded area represents the deviation of the five runs.

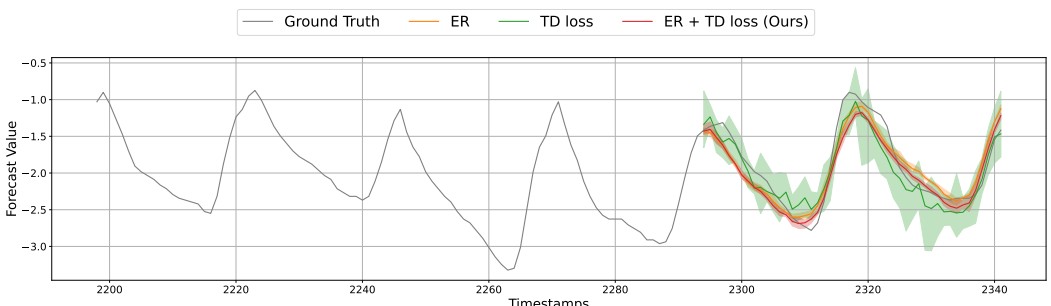

(a) Sequences with predictable cycles: ER offers more stable and confident predictions while TD loss exhibit large variance.

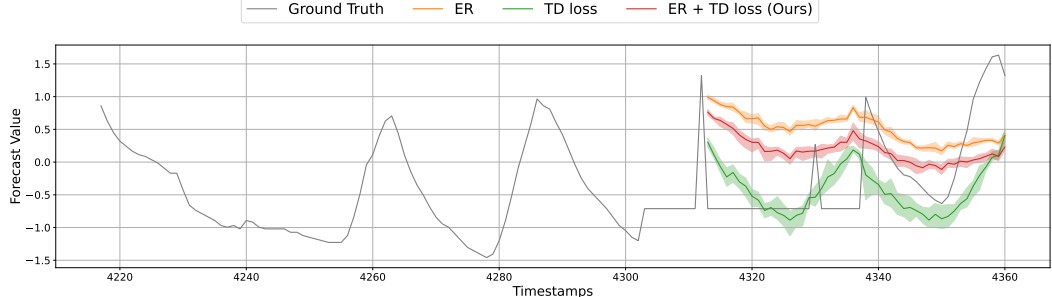

(b) Sequences with sudden changes: ER may form outdated assumptions about the current distribution while TD adapts to the shift.

Figure 5: Visualization of the interaction between ER and TD loss components in ETTh2.

By training the model with batches of historical sequences, the ER component allows for more assured predictions, especially when similar patterns are encountered. In contrast, the TD loss component relies on pseudo labels defined in Equation 5 for updates, where only the initial data

point is based on ground truth, and the remaining points are less certain predictions from the teacher model. This approach results in high sensitivity to minor fluctuations in sequences, causing greater variance in predictions. As illustrated in Figure 5a, relying solely on TD loss can result in significant variance in predictions, even for predictable sequences with regular cycles. In these cases, ER helps reduce variance by offering more stable and confident predictions.

For immediate changes in data distribution, TD loss can make rapid adaptations. As shown in Figure 5b, it effectively identifies sudden changes around timestamp 4300, leading to lower error in predictions. In comparison, the ER component, reliant on historical data, may make outdated assumptions about the current distribution. It may make confident predictions that are inaccurate in rapidly changing environments.

The interaction between ER and TD loss resembles a bias-variance trade-off. By utilizing both components, the model can harness their strengths to balance this trade-off, resulting in more accurate and reliable predictions across different data distributions.

### A.2.4 RUNTIME EFFICIENCY ANALYSIS OF THE COMPONENTS

We measure the time taken for online training per sample on the two main components of the network, ER and TD loss. Consistent with Section A.1, the wall time statistics in Table 8 and processor time statistics in Table 9 are based on 250 repeated runs, using ms/itr as the unit of measurement, where a lower number indicates a shorter runtime. Note that a replay batch size of 32 was utilized in this experiment. The results clearly show that ER contributes more to runtime than TD loss. A potential future direction is to develop a method with lower runtime complexity to replace the current ER strategy, or use a smaller replay batch size as in Section A.5.2. This is particularly relevant for datasets with a very high feature dimension such as Traffic.

Table 8: Wall time comparison for the two main components of the framework, measured in ms/itr. Lower values indicate higher efficiency.

| Teacher Model | Student Model | ER | TD | ECL | | ETTh2 | | Traffic | |
|---|---|---|---|---|---|---|---|---|---|
| | | | | Mean | SD | Mean | SD | Mean | SD |
| DLinear | MLP | ✗ | ✗ | 5.58 | 0.18 | 4.63 | 0.14 | 11.03 | 1.06 |
| | | ✓ | ✗ | 27.98 | 2.18 | 8.12 | 0.10 | 43.74 | 3.45 |
| | | ✗ | ✓ | 8.32 | 0.51 | 6.53 | 0.44 | 17.85 | 1.60 |
| | | ✓ | ✓ | 30.76 | 1.06 | 12.04 | 0.16 | 63.96 | 12.42 |

Table 9: Processor time comparison for the two main components of the framework, measured in ms/itr. Lower values indicate higher efficiency.

| Teacher Model | Student Model | ER | TD | ECL | | ETTh2 | | Traffic | |
|---|---|---|---|---|---|---|---|---|---|
| | | | | Mean | SD | Mean | SD | Mean | SD |
| DLinear | MLP | ✗ | ✗ | 4.90 | 0.36 | 3.67 | 0.18 | 166.87 | 19.09 |
| | | ✓ | ✗ | 434.26 | 16.37 | 7.58 | 0.68 | 637.02 | 19.47 |
| | | ✗ | ✓ | 7.00 | 0.22 | 5.30 | 0.05 | 289.61 | 27.26 |
| | | ✓ | ✓ | 478.45 | 31.96 | 10.91 | 1.20 | 800.70 | 10.74 |

### A.3 FREEZING STRATEGIES OF TEACHER AND STUDENT MODELS

The effects of three freezing settings are examined in Table 10: freezing both the teacher and student models (i.e., batch learning), freezing only the teacher model, and not freezing either model. Results indicated that updating both models yields the best performance, followed by updating only the student model. Existing forecasting methods learned through batch learning may struggle to generalize future distribution shifts due to the continuous emergence of new patterns. Integrating these methods into an online learning framework can improve adaptability and performance in dynamic environments.

Table 10: MSE comparison across different freezing strategies for teacher and student models. Three settings are analyzed: freezing both the teacher and student models (batch learning), freezing only the teacher model, and not freezing either model. All $\theta^{(S)}$ denotes the parameters of MLP.

| Teacher Model | $\theta^{(T)}$ | $\theta^{(S)}$ | ECL | | | ETTh2 | | | Traffic | | |
|---|---|---|---|---|---|---|---|---|---|---|---|
| | | | 1 | 24 | 48 | 1 | 24 | 48 | 1 | 24 | 48 |
| FSNet | ❄️ | ❄️ | 417.3 | 478.1 | 485.2 | 11.84 | 21.83 | 19.3 | 0.592 | 0.749 | 0.789 |
| | ❄️ | 🔥 | 153.1 | 396 | 411.3 | 5.944 | 8.558 | 9.014 | **0.267** | 0.518 | 0.580 |
| | 🔥 | 🔥 | **2.233** | **5.446** | **6.475** | **0.669** | **3.392** | **5.188** | 0.334 | **0.420** | **0.398** |

## A.4 LEARNING STRATEGIES OF STUDENT MODEL

The use of a student model learning strategy is investigated in our method. By employing a residual learning strategy, the student model utilizes the predictions of the teacher model along with the horizon window as input, subsequently fine-tuning the teacher's predictions, as illustrated in Equation 4. In contrast, the separate learning strategy provides the student model only the horizon window, and its predictions are directly used for evaluation, independent of the teacher model's prediction. A comparison of the performance of our method (i.e., residual learning) and separate learning is conducted, with the results presented in Table 11.

Table 11: Comparison of student model learning strategies: Residual learning vs. separate learning. In the "Residual" column, a cross ("✗") indicates separate learning, where the student model relies solely on the horizon window, using its predictions as the sole output for evaluation. Conversely, a tick ("✓") indicates residual learning, enabling the student model to refine the teacher's predictions by utilizing both the teacher's output and the horizon window.

| | Student Model | Residual | ECL | | ETTh2 | | Traffic | |
|---|---|---|---|---|---|---|---|---|
| | | | 24 | 48 | 24 | 48 | 24 | 48 |
| FSNet | MLP | ✗ | 181.7 | 193.5 | 8.743 | 9.828 | 0.872 | 0.894 |
| | | ✓ | **5.446** | **6.475** | **3.392** | **5.188** | 0.420 | **0.398** |
| | FSNet | ✗ | 39.31 | 40.66 | 3.496 | 6.858 | 0.388 | 0.417 |
| | | ✓ | 8.242 | 12.67 | 3.435 | 5.419 | 0.407 | 0.444 |
| | PatchTST | ✗ | 6.321 | 7.065 | 11.74 | 12.15 | 0.451 | 0.490 |
| | | ✓ | 6.93 | 7.283 | 3.461 | 5.761 | **0.367** | 0.401 |
| PatchTST | MLP | ✗ | 5.476 | 6.765 | 2.39 | 3.719 | 0.578 | 0.666 |
| | | ✓ | **4.962** | **6.113** | **1.979** | 3.905 | 0.419 | 0.447 |
| | FSNet | ✗ | 79.59 | 47.52 | 2.703 | 4.953 | **0.388** | **0.420** |
| | | ✓ | 8.633 | 10.52 | 2.248 | 4.215 | 0.411 | 0.443 |
| | PatchTST | ✗ | 5.874 | 6.82 | 11.62 | 12.95 | 0.445 | 0.483 |
| | | ✓ | 24.11 | 53.55 | 1.996 | **3.52** | 0.480 | 0.473 |

Two observations can be made:

1. Residual training enables simpler models, reducing computational costs and memory usage. Using an MLP as the student model in residual learning outperforms more complex models trained with separate strategies. This is likely because correcting distribution shift errors is easier than mastering the main task.

2. When choosing a student model for residual learning, a simpler model is preferred. Replacing the MLP with the more complex FSNet shows similar performance, but using PatchTST leads to deterioration. This may be due to PatchTST's complexity, making it ill-suited for the simpler task of learning distribution shift errors. Thus, when using PatchTST, a separate learning approach is more effective.

## A.5 Key Elements of Experience Replay: Buffer Size and Update Frequency

### A.5.1 Buffer Size

We investigate the impact of replay buffer size on our method's performance across three datasets (ECL, ETTh2, and Traffic) using various teacher model backbones (DLinear, FSNet, and PatchTST) and an MLP as the student model. As shown in Figure 6, all plots exhibit an elbow-shaped pattern, indicating rapid performance increases with initial buffer size increments. However, beyond a size of 300, the improvement in performance becomes marginal. It is observed that, in general, a larger replay buffer size leads to better performance. However, when the buffer size exceeds 300, the improvements become marginal. Generally, larger replay buffer sizes enhance performance, but excessively large buffers may lead to declines due to training on outdated or out-of-distribution data.

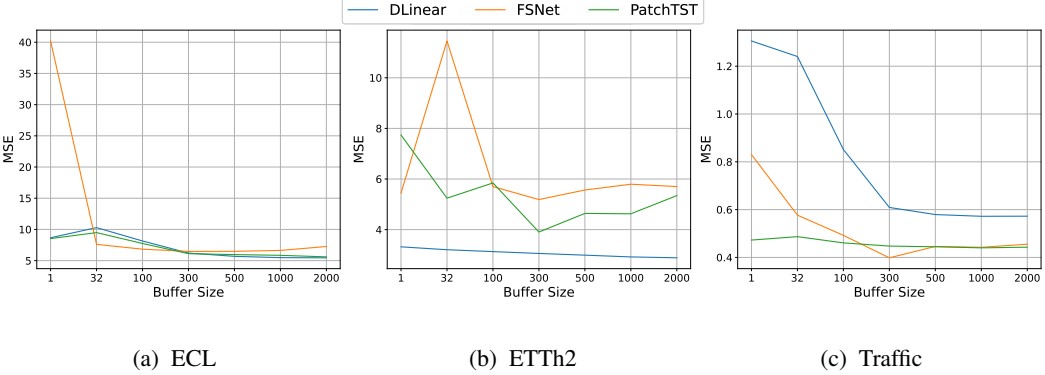

|            (a) ECL            |           (b) ETTh2           |           (c) Traffic          |

Figure 6: Performance analysis of varying replay buffer sizes, using DLinear, FSNet, and PatchTST as teacher models, with MLP as the student model.

Table 12: MSE comparison across different batch sizes in ER.

| Teacher Model | Student Model | Batch Size | ECL 1 | ECL 24 | ECL 48 | ETTh2 1 | ETTh2 24 | ETTh2 48 | Traffic 1 | Traffic 24 | Traffic 48 |
|---|---|---|---|---|---|---|---|---|---|---|---|
| DLinear | MLP | 8 | 2.104 | 4.803 | 6.433 | 0.373 | 1.744 | 3.091 | 0.299 | 0.600 | 0.685 |
|  |  | 16 | 2.075 | 4.771 | 6.364 | 0.368 | 1.715 | 3.072 | 0.300 | 0.602 | 0.683 |
|  |  | 32 | 2.065 | 4.737 | 6.181 | 0.365 | 1.701 | 3.082 | 0.302 | 0.608 | 0.681 |
|  |  | 64 | 2.047 | 4.758 | 6.085 | 0.367 | 1.682 | 3.116 | 0.303 | 0.609 | 0.681 |
| FSNet | MLP | 8 | 2.200 | 5.409 | 6.763 | 0.476 | 3.297 | 5.555 | 0.223 | 0.367 | 0.403 |
|  |  | 16 | 2.144 | 5.401 | 6.916 | 0.562 | 3.467 | 5.795 | 0.224 | 0.364 | 0.400 |
|  |  | 32 | 2.330 | 5.475 | 7.000 | 0.431 | 3.114 | 5.318 | 0.228 | 0.366 | 0.403 |
|  |  | 64 | 2.482 | 5.513 | 7.225 | 0.603 | 3.063 | 5.784 | 0.235 | 0.370 | 0.409 |
| PatchTST | MLP | 8 | 2.680 | 5.275 | 6.318 | 0.383 | 1.884 | 3.819 | 0.231 | 0.436 | 0.460 |
|  |  | 16 | 2.583 | 5.109 | 6.173 | 0.373 | 1.872 | 3.900 | 0.231 | 0.435 | 0.455 |
|  |  | 32 | 2.244 | 5.169 | 6.665 | 0.382 | 1.925 | 4.473 | 0.232 | 0.432 | 0.454 |
|  |  | 64 | 2.382 | 5.150 | 6.978 | 0.381 | 1.976 | 4.832 | 0.235 | 0.433 | 0.455 |

### A.5.2 Replay Batch Size

The impact of the replay buffer batch size on model performance is explored by testing batch sizes of 8, 16, 32, and 64. As noted in Section A.2.4, ER tends to occupy long runtime. This led us to consider the trade-offs of modifying the batch size, aiming to pinpoint the scenarios where increasing the batch size would result in meaningful performance gains that justify the extra computational expense. By identifying these scenarios, we can make informed choices about when it is beneficial

to choose larger batch sizes, weighing the improved performance against the downside of extended runtime.

The MSE results are presented in Table 12. Overall, a batch size of 16 is more effective for the ECL dataset, while the ETTh2 dataset benefits from a smaller batch size of 8, especially for a prediction length of 48. For the Traffic dataset, no specific pattern emerges, and the results remain relatively consistent across different batch sizes. Given the lack of significant performance gains with a larger batch size like 64, it is recommended maintaining a smaller batch size of 8 or 16 in ER to minimize the trade-off between efficiency and accuracy.

### A.5.3 MODEL UPDATE FREQUENCY WITH ER

The impact of update frequency in ER on method performance is investigated. Performance comparisons are made using various update frequencies, as shown in Table 13. Whole numbers indicate more frequent updates, while fractions denote less frequent updates. For instance, "3" signifies three updates using the ER method, whereas "1/3" indicates one update for every three data streams.

Table 13: Impact of ER update frequency on method performance. In the "Freq" column, whole numbers indicate more frequent updates, while fractions represent less frequent updates. For example, a "3" in that column signifies updating the model three times using the ER method, while "1/3" implies updating the model once for every three data streams that arrive.

| Teacher Model | Student Model | Freq | ECL | | | ETTh2 | | | Traffic | | |
|---|---|---|---|---|---|---|---|---|---|---|---|
| | | | 1 | 24 | 48 | 1 | 24 | 48 | 1 | 24 | 48 |
| DLinear | MLP | 1/3 | 2.066 | 4.769 | 6.892 | 0.373 | 1.849 | 3.302 | 0.246 | 0.514 | 0.563 |
| | | 1/2 | 2.072 | 4.75 | 6.562 | 0.370 | 1.78 | 3.187 | 0.244 | 0.516 | 0.562 |
| | | 1 | 2.07 | 4.767 | 6.21 | 0.378 | 1.732 | 3.058 | 0.243 | 0.546 | 0.609 |
| | | 2 | 2.074 | 4.799 | 6.035 | 0.362 | 1.704 | 3.088 | 0.250 | 0.521 | 0.559 |
| | | 3 | 2.09 | 4.814 | 6.085 | 0.362 | 1.699 | 3.116 | 0.248 | 0.532 | 0.567 |
| FSNet | MLP | 1/3 | 2.425 | 5.829 | 7.025 | 0.555 | 3.227 | 5.942 | 0.334 | 0.406 | 0.445 |
| | | 1/2 | 2.384 | 5.623 | 7.075 | 0.629 | 3.415 | 6.1 | 0.320 | 0.411 | 0.443 |
| | | 1 | 2.233 | 5.446 | 6.475 | 0.669 | 3.392 | 5.188 | 0.334 | 0.420 | 0.398 |
| | | 2 | 2.389 | 5.469 | 6.833 | 0.614 | 2.955 | 5.311 | 0.325 | 0.417 | 0.464 |
| | | 3 | 2.334 | 5.468 | 6.783 | 0.711 | 3.081 | 5.889 | 0.316 | 0.423 | 0.471 |
| PatchTST | MLP | 1/3 | 2.562 | 4.782 | 6.4 | 0.470 | 3.349 | 9.962 | 0.236 | 0.425 | 0.456 |
| | | 1/2 | 2.409 | 4.852 | 6.291 | 0.415 | 2.622 | 8.649 | 0.235 | 0.422 | 0.453 |
| | | 1 | 2.357 | 4.962 | 6.113 | 0.400 | 1.979 | 3.905 | 0.231 | 0.419 | 0.447 |
| | | 2 | 2.211 | 5.193 | 6.761 | 0.383 | 1.859 | 4.214 | 0.229 | 0.418 | 0.446 |
| | | 3 | 2.175 | 5.167 | 7.152 | 0.378 | 1.803 | 4.438 | 0.230 | 0.418 | 0.447 |

Contrary to the common belief, a higher frequency of ER updates may not necessarily lead to improved performance. As an example, Figure 7a shows four distinct temporal distributions from the Electricity dataset: the first spans from time step 1600 to 1660, the second from 1660 to 1715, the third from 1715 to 1830, and the fourth from 1830 to 1875. Following the distributions transitions at steps 1660, 1715, and 1830, a lower frequency of ER updates is associated with better performance, as shown in Figure 7b, where the MSE values are smaller at these time steps. This may be attributable to the model's increased focus on TD loss when ER is less frequent, allowing for greater emphasis on immediate ground truth over more distant data streams.

On the other hand, a lower frequency of ER updates limits the model's adaptability to the new ongoing distribution after a transition. Between steps 1775 and 1830, as depicted in Figure 7a, lower ER frequency leads to greater deviations from the ground truth. Meanwhile, as shown in Figure 7b, high MSE values persist for a longer duration. This decline can be attributed to insufficient training on in-distribution samples, which hampers generalization to near-future data. Therefore, it is advisable to strike a balance between learning from ER and TD loss, suggesting a moderate ER update frequency, such as updating once for each data stream.

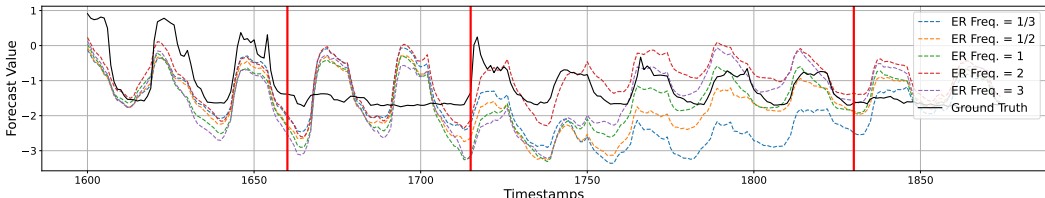

(a) Prediction results based on different ER update frequencies.

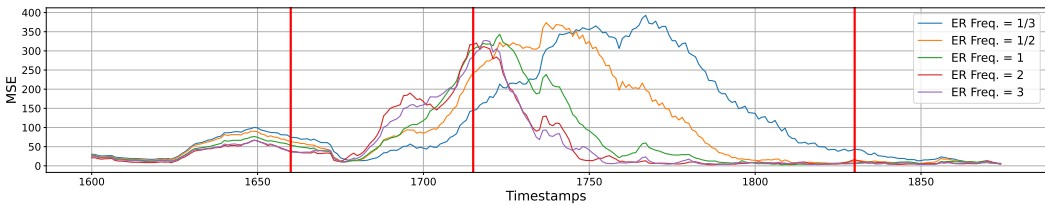

(b) MSE loss at individual timestamps.

Figure 7: Impact of ER update frequency on model performance across four temporal distributions from the Electricity dataset: 1600-1660, 1660-1715, 1715-1830, and 1830-1875.

## A.6 USE OF TEMPORAL DIFFERENCE LOSS AND ITS REGULARIZATION EFFECTS

### A.6.1 REGULARIZATION EFFECTS

The impact of the TD loss, as described in Equation 6, and its regularization factor on forecasting performance is examined using three datasets (ECL, ETTh2, ETTm1) with FSNet. Regularization is controlled by adjusting the parameter $\gamma$ of the TD loss. In Figure 8, $\gamma = 0$ indicates that the TD loss is not used, and the model updates solely with ER during the online phase. In contrast, for $\gamma = 1$, the model updates with the fast data stream using TD loss without any regularization.

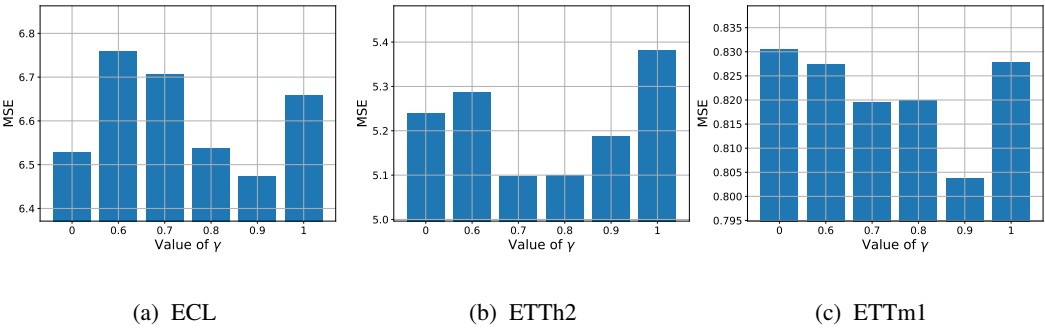

(a) ECL          (b) ETTh2          (c) ETTm1

Figure 8: MSE results for various TD loss parameter choices. The parameter $\gamma = 0$ signifies that the TD loss is not utilized, meaning the model updates solely based on ER during testing. In contrast, when $\gamma = 1$, the model updates with the fast data stream without any regularization applied.

Figure 8 illustrates that the best results are not achieved with $\gamma = 0$ or $\gamma = 1$. Optimal performance is attained with a specific degree of regularization. However, a drawback of our method is its sensitivity to the choice of the regularization factor. For instance, in Figure 8a, failing to select $\gamma = 0.9$ may result in worse performance than either not using the TD loss or not using any regularization at all.

### A.6.2 ENHANCING PSEUDO LABEL CONSTRUCTION

Extending the method of constructing a pseudo label as described in Equation 5, a new parameter $k$ is introduced in Equation 8. This parameter allows for adjusting the balance between ground truth

data and the teacher model's predictions. The value $k$ indicates the use of the most recent $k$ ground truth data points and the first $H - k$ steps of teacher model's prediction to construct the pseudo label.

$$\tilde{X}^{(i)}_{i-k+1:i+H-k} = [X_{i-k+1:i}, \hat{X}^{(T,i)}_{i+1,i+H-k}] \tag{8}$$

Using $k = 1$ makes the pseudo label unreliable due to limited ground truth data. We aim to explore whether the student model can enhance its learning effectiveness as the value of $k$ increases, or by utilizing multiple values of $k$ simultaneously.

**Altering the values of $k$.** In this experiment, our aim is to find the best ratio of ground truth values to the teacher model's predictions for constructing the pseudo label. Setting $H = 48$ and using DLinear as the teacher model of DSOF, we conduct the experiment by selecting a single value of $k$ instead of multiple values for the fast stream, and testing with evenly distributed $k$ values. Specifically, we chose multiples of 4, i.e. $k \in \{1, 4, 8, 12, \ldots, 40, 44\}$. As shown in Figure 9, smaller values of $k$ are preferred. The results align with our expectations, indicating that the model should be updated more promptly with the most recent data points.

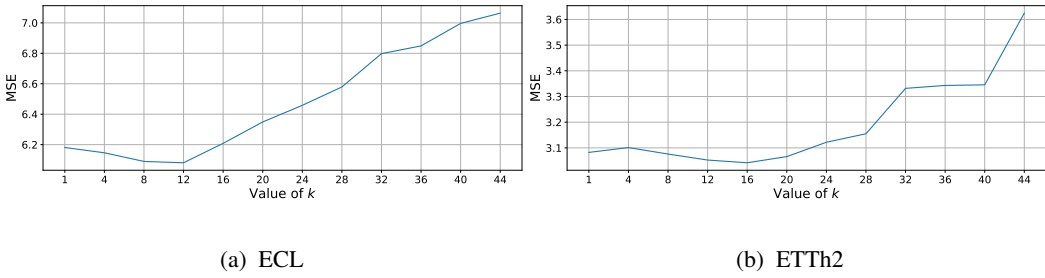

(a) ECL          (b) ETTh2

Figure 9: MSE analysis for different values of $k$ in pseudo label construction. The value $k$ indicates the use of the most recent $k$ ground truth data points and the first $H - k$ steps of teacher model's prediction for pseudo label construction.

**Using Multiple Values of $k$ Simultaneously.** In this experiment, instead of relying solely on the most recent data, we utilize a range of $k$ values, specifically powers of 2 that are less than the prediction sequence length. Specifically, at $t = i$, the fast data stream uses the batch of pseudo labels $\{\tilde{X}^{(i)}_{i-k+1:i+H-k} | k \in \mathcal{K}\}$, where $\mathcal{K} = \{2^p | 2^p \le H, p \in \mathbb{N}\}$, to update the student model. For example, if $H = 24$ and $t = 20$, the fast data stream uses $k \in \{1, 2, 4, 8, 16\}$, resulting in the batch $[\tilde{X}^{(20)}_{20:43}, \tilde{X}^{(20)}_{19:42}, \tilde{X}^{(20)}_{17:40}, \tilde{X}^{(20)}_{13:36}, \tilde{X}^{(20)}_{5:28}]$.

Table 14: MSE comparison of between using a single $k$ value ($k = 1$) and utilizing multiple $k$ values concurrently ($k = 2^p$).

| | | DLinear | | FSNet | | PatchTST | |
|---|---|---|---|---|---|---|---|
| Datasets | H | $k = 1$ | $k = 2^p$ | $k = 1$ | $k = 2^p$ | $k = 1$ | $k = 2^p$ |
| | 24 | **4.737** | 4.803 | 5.475 | **5.323** | **5.169** | 5.191 |
| Electricity | 48 | 6.181 | **6.022** | 7.000 | **6.821** | 6.665 | **6.566** |
| | 96 | 10.661 | **10.393** | 10.023 | **9.839** | 11.248 | **10.249** |
| | 24 | **1.701** | 1.713 | 3.114 | **2.742** | 1.925 | **1.874** |
| ETTh2 | 48 | 3.082 | **3.015** | 5.318 | **4.959** | 4.473 | **3.970** |
| | 96 | 5.609 | **5.323** | 9.526 | **9.446** | 17.363 | **13.351** |

In Table 14, the approach of using powers of 2 is denoted as $k = 2^p$. The results indicate that this method is particularly beneficial for larger values of $H$. For instance, applying to the ETTh2 dataset using PatchTST with $H = 96$ provides significant improvements.

## A.7 Learning Rate Sensitivity Analysis

This section explores the sensitivity of the learning rate in our method. The online learning rate $\alpha_O$ in Algorithm 1 is varied while keeping the batch learning rates constant. Performance comparisons are made with learning rates ranging from 0.0001 to 0.003 at five intervals using four teacher model backbones, DLinear, FSNet, PatchTST, and iTransformer, on ECL, as shown in Figure 10.

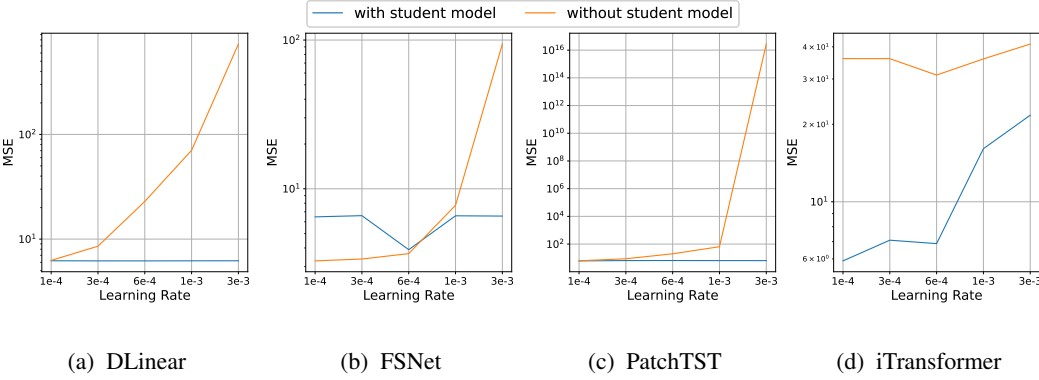

(a) DLinear      (b) FSNet      (c) PatchTST      (d) iTransformer

Figure 10: Sensitivity analysis of the learning rate in our method, highlighting performance across various online learning rates on the ECL dataset. By default, the student model is an MLP. In cases where the student model is not used, the teacher model is continuously updated with ER and TD loss, without separate forecasting heads.

We conduct comparisons under two different settings. In the first setting, similar to previous experiments, by default, we use an MLP as the student model when one is employed. In the second setting, when a student model is not used, the teacher model is continuously updated with ER and TD loss without separate forecasting heads. During fast data stream updates, the teacher model adjusts its parameters based on its predictions compared to pseudo labels, rather than freezing its training and backpropagating it to the student model.

It is observed that training transformer backbones like iTransformer and PatchTST in an online setting without a student model is more challenging and makes it harder to reach convergence. This is particularly true for datasets like ECL, which exhibit significant fluctuations and large values, even after standardization in data preprocessing. In such cases, a student model enhances robustness to the learning rate. However, for non-transformer models like DLinear and FSNet, the teacher-student model may be unnecessary. While the teacher-student model approach shows robustness across various learning rates, a sufficiently small learning rate (e.g., less than 0.0001) can still yield performance comparable to using only the teacher model.

## A.8 Impact of Look-back Window Length

In this section, we explore how the length of the look-back window impacts the forecasting performance of various models. We consider the look-back window length as a hyperparameter and perform a grid search over values $L \in \{24, 48, 96, 192\}$ using selected datasets, models, and forecasting horizons.

Contrary to the common belief that longer look-back windows provide more information and thus enhance forecasting performance, our findings suggest otherwise. The results shown in Table 15 reveal that, in most cases, model performance declines as the look-back window length increases. Interestingly, a look-back window length of 24 often results in better performance, regardless of the horizon window. This may be due to the ER strategy used. Recall that the slow stream must wait for $L+H$ steps to ensure it only includes data pairs for which all ground truth values have been received. With larger $L$, the data sequences have to wait longer before being added to the buffer, which might lead to a less informative slow stream and reduce the effectiveness of the ER procedure.

Table 15: Impact of look-back window length $L \in \{24, 48, 96, 192\}$ on MSE results across datasets and forecasting horizons $H$.

| Datasets | H/L | DLinear | | | | iTransformer | | | | PatchTST | | | |
|---|---|---|---|---|---|---|---|---|---|---|---|---|---|
| | | 24 | 48 | 96 | 192 | 24 | 48 | 96 | 192 | 24 | 48 | 96 | 192 |
| ECL | 1 | 2.802 | 2.258 | 2.065 | **2.010** | 2.341 | **2.282** | 2.430 | 2.349 | 2.603 | 2.265 | **2.244** | 2.476 |
| | 24 | 5.396 | 4.845 | **4.737** | 4.980 | 5.651 | **5.101** | 5.155 | 5.571 | 5.247 | **5.128** | 5.169 | 5.516 |
| | 48 | 6.563 | 6.270 | **6.181** | 6.778 | 6.736 | 6.324 | **6.015** | 7.734 | **6.333** | 6.366 | 6.665 | 8.461 |
| ETTh2 | 1 | **0.364** | 0.369 | 0.365 | 0.371 | **0.371** | 0.398 | 0.384 | 0.374 | 0.385 | 0.388 | **0.382** | 0.394 |
| | 24 | **1.664** | 1.666 | 1.701 | 1.891 | **1.658** | 1.704 | 1.869 | 2.103 | 1.859 | **1.850** | 1.925 | 2.135 |
| | 48 | **2.839** | 2.951 | 3.082 | 3.552 | **2.801** | 2.972 | 3.465 | 4.035 | 3.506 | **3.244** | 4.473 | 5.259 |
| Ex. | 1 | **0.009** | 0.009 | 0.009 | 0.010 | **0.009** | 0.011 | 0.010 | 0.014 | 0.010 | **0.009** | 0.011 | 0.010 |
| | 24 | **0.092** | 0.093 | 0.095 | 0.101 | **0.095** | 0.100 | 0.110 | 0.115 | **0.093** | 0.101 | 0.103 | 0.107 |
| | 48 | **0.184** | 0.186 | 0.192 | 0.205 | **0.206** | 0.228 | 0.218 | 0.305 | **0.188** | 0.202 | 0.213 | 0.242 |

# B CONNECTING TEMPORAL DIFFERENCE LEARNING IN REINFORCEMENT LEARNING TO DSOF

As outlined in Section 3.3.2, the fast stream leverages ideas from TD learning to quickly process latest data and improve near-future forecasts. While the learning goals of RL and OTSF differ, both fields share the challenge of updating models without complete access of ground truth information. In this section, drawing inspiration from TD learning in RL, we demonstrate how the idea of estimating the value of the next state can be applied to facilitate intermediate updates in OTSF.

Recall the definition of temporal difference learning in the context of reinforcement learning (Sutton & Barto, 1999) as given in Equation 9, where $V(S_i)$ is the value of state $S$ at time $i$, $\alpha$ is the learning rate, $R_{i+1}$ is the reward given at $i + 1$ and $\gamma V(S_{i+1})$ is the discounted value of the next state.

$$V(S_i) \leftarrow V(S_i) + \alpha[R_{i+1} + \gamma V(S_{i+1}) - V(S_i)] \tag{9}$$

We draw a connection between our scenario and TD learning in reinforcement learning in the following way:

- The agent corresponds to the student model.
- The state $S_{i-1}$ corresponds to the teacher model's prediction output $\hat{X}_{i:i+H-1}^{(T,i-1)}$ at time $i-1$.
- The action corresponds to the student model's prediction output $\hat{X}_{i:i+H-1}^{(S,i-1)}$ at time $i - 1$, based on the "state" $\hat{X}_{i:i+H-1}^{(T,i-1)}$.
- The value $V(S_{i-1})$ corresponds to the prediction output of the teacher-student model $\hat{X}_{i:i+H-1}^{(i-1)}$ at time $i - 1$.
- The reward $R_i$ corresponds to the negative MSE between the first prediction point and the ground truth $x_i$, once it becomes available at time $i$.
- The discounted value of the next state, $\gamma V(S_i)$, corresponds to the importance of the teacher model's prediction output at time $i$, $\hat{X}_{i+1:i+H}^{(T,i)}$, relative to the teacher-student model's prediction at time $i - 1$, $\hat{X}_{i:i+H-1}^{(i-1)}$. In reinforcement learning, the discounted factor is essential because the agent faces a probability of failure or uncertainty during exploration at each time step. This is similar to our OTSF scenario, where predictions for data points further from the current observations become increasingly unreliable.

While insights from TD learning in RL can potentially enhance OTSF performance, OTSF should not be framed as an RL problem because their primary objectives are fundamentally different. RL focuses on optimizing sequence of actions for maximizing long-term rewards, whereas OTSF seeks accurate near-future predictions for data streams which may have changing distributions. This distinction leads to unique algorithmic settings; for instance, in RL, the current action influences the next state, whereas in OTSF, previous forecasts do not impact subsequent ones.

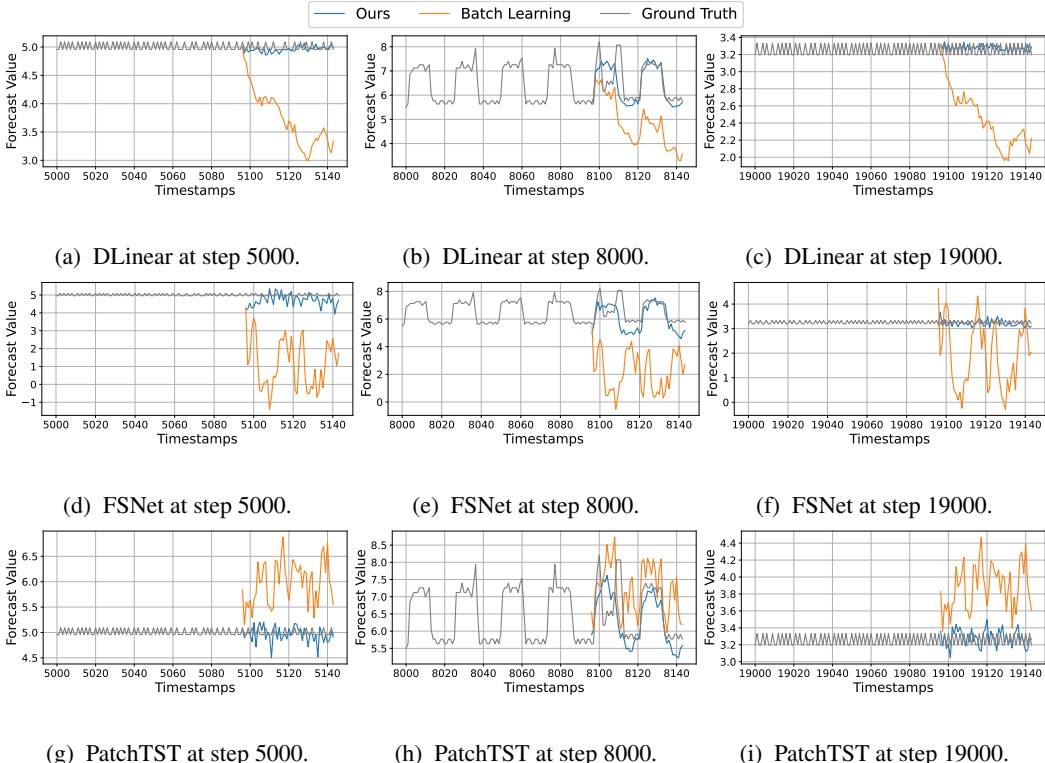

Figure 11: Visualization of the online learning process for ECL at steps 5000, 8000 and 19000.

## C FORMULATION OF INFORMATION LEAKAGE IN EXISING ONLINE LEARNING FRAMEWORKS

Information leakage occurs when data points previously used for model parameter updates are also used for current evaluation.

In existing online learning frameworks, at time $t = i$, the model first takes in input sequence from $t = i - L - H + 2$ to $t = i - H + 1$, then predicts values from $t = i - H + 2$ to $t = i + 1$.

At $t = i+1$, the ground truth at $t = i+1$ becomes available, and the most recent prediction made at $t = i$ is utilized for evaluation. The model then updates its parameters using the ground truth from $t = i - H + 2$ to $t = i + 1$ and predicts values from $t = i - H + 3$ to $t = i + 2$.

At $t = i + 2$, ground truth at $t = i + 2$ is available. The last prediction made at $t = i + 1$, with timestamps from $t = i - H + 3$ to $t = i + 2$, is being used for evaluation. Information leak occurs here because ground truth with timestamps from $t = i - H + 3$ to $t = i + 1$ was already used for parameter updates at $t = i + 1$.

## D ADDITIONAL VISUALIZATIONS

In this section, we offer additional insights into the prediction performance of DSOF in comparison to batch learning, illustrating how it addresses various time series patterns across different datasets. In addition to examples from the Traffic dataset in Figure 3, we also present examples from the Electricity and ETTh2 datasets in Figure 11 and Figure 12, respectively.

Table 16: Detailed comparison of MSE results between our framework and batch learning across various datasets and backbones. The better results from the two settings are highlighted in bold.

| | | DLinear | | | | FITS | | | | FSNet | | | | OneNet | | | | iTrans. | | | | PatchTST | | | | NSTrans. | | | |
|---|---|---|---|---|---|---|---|---|---|---|---|---|---|---|---|---|---|---|---|---|---|---|---|---|---|---|---|---|---|
| | H | Batch Learn. | SD | DSOF | SD | Batch Learn. | SD | DSOF | SD | Batch Learn. | SD | DSOF | SD | Batch Learn. | SD | DSOF | SD | Batch Learn. | SD | DSOF | SD | Batch Learn. | SD | DSOF | SD | Batch Learn. | SD | DSOF | SD |
| ECL | 1 | 2.842 | 0.436 | **2.065** | 0.004 | 2.870 | 0.036 | **2.235** | 0.008 | 3.8e+2 | 1.5e+1 | **2.330** | 0.056 | 3.2e+1 | 1.6e+1 | **4.733** | 0.746 | **1.976** | 0.036 | 2.430 | 0.085 | 4.770 | 1.773 | **2.244** | 0.168 | 3.9e+1 | 3.258 | **2.703** | 0.066 |
| | 24 | 1.5e+1 | 3.024 | **4.737** | 0.018 | **4.509** | 0.002 | 4.597 | 0.009 | 4.7e+2 | 2.816 | **5.475** | 0.043 | 8.5e+1 | 8.561 | **4.510** | 0.157 | **4.119** | 0.014 | 5.155 | 0.039 | 1.7e+1 | 6.272 | **5.169** | 0.053 | 4.0e+1 | 5.944 | **8.668** | 0.612 |
| | 48 | 2.7e+1 | 7.014 | **6.181** | 0.021 | **5.257** | 0.009 | 5.433 | 0.012 | 4.8e+2 | 3.927 | **7.000** | 0.133 | 1.5e+2 | 1.8e+1 | **5.943** | 0.090 | **4.936** | 0.070 | 6.015 | 0.027 | 1.7e+1 | 5.385 | **6.665** | 0.193 | 3.8e+1 | 3.286 | **9.981** | 1.076 |
| ETTh2 | 1 | 0.470 | 0.011 | **0.365** | 0.004 | 0.522 | 0.015 | **0.375** | 0.004 | 1.1e+1 | 1.772 | **0.431** | 0.012 | 2.531 | 0.166 | **0.548** | 0.048 | 0.872 | 0.118 | **0.384** | 0.016 | 0.915 | 0.528 | **0.382** | 0.008 | 3.088 | 0.104 | **0.415** | 0.017 |
| | 24 | 2.269 | 0.075 | **1.701** | 0.005 | 2.189 | 0.031 | **1.757** | 0.010 | 1.9e+1 | 1.693 | **3.114** | 0.089 | 7.017 | 0.966 | **2.363** | 0.087 | 2.688 | 0.152 | **1.869** | 0.015 | 5.213 | 1.036 | **1.925** | 0.058 | 3.973 | 0.030 | **2.127** | 0.075 |
| | 48 | 3.389 | 0.097 | **3.082** | 0.027 | 3.275 | 0.019 | **2.988** | 0.010 | 2.3e+1 | 2.381 | **5.318** | 0.175 | 9.790 | 1.577 | **4.037** | 0.245 | 3.769 | 0.229 | **3.465** | 0.066 | 6.566 | 0.902 | **4.473** | 0.729 | 4.840 | 0.102 | **4.729** | 0.603 |
| ETTm1 | 1 | 0.112 | 0.003 | **0.105** | 0.000 | 0.123 | 0.003 | **0.111** | 0.000 | 0.190 | 0.019 | **0.121** | 0.008 | 0.156 | 0.019 | **0.096** | 0.008 | 0.179 | 0.027 | **0.152** | 0.002 | 0.155 | 0.015 | **0.108** | 0.001 | 1.101 | 0.290 | **0.141** | 0.004 |
| | 24 | 0.628 | 0.016 | **0.525** | 0.000 | 0.732 | 0.009 | **0.542** | 0.001 | 1.500 | 0.163 | **0.609** | 0.011 | 1.094 | 0.056 | **0.418** | 0.035 | 1.049 | 0.106 | **0.618** | 0.010 | 1.170 | 0.119 | **0.582** | 0.006 | 1.503 | 0.006 | **0.904** | 0.051 |
| | 48 | 0.818 | 0.022 | **0.695** | 0.001 | 0.900 | 0.009 | **0.716** | 0.002 | 2.279 | 0.268 | **0.843** | 0.017 | 1.588 | 0.120 | **0.554** | 0.024 | 1.320 | 0.096 | **0.856** | 0.019 | 2.103 | 0.279 | **0.805** | 0.011 | 1.547 | 0.003 | **1.119** | 0.087 |
| Ex. | 1 | **0.009** | 0.000 | **0.009** | 0.000 | 0.012 | 0.000 | **0.011** | 0.001 | 0.034 | 0.008 | **0.010** | 0.000 | 0.024 | 0.004 | **0.009** | 0.000 | 0.011 | 0.001 | **0.010** | 0.000 | 0.024 | 0.015 | **0.011** | 0.002 | 0.234 | 0.023 | **0.015** | 0.001 |
| | 24 | 0.098 | 0.005 | **0.095** | 0.000 | 0.095 | 0.000 | **0.093** | 0.000 | 0.754 | 0.230 | **0.120** | 0.002 | 0.487 | 0.079 | **0.152** | 0.009 | 0.117 | 0.004 | **0.110** | 0.005 | 0.443 | 0.111 | **0.103** | 0.001 | 0.335 | 0.013 | **0.127** | 0.007 |
| | 48 | 0.194 | 0.014 | **0.192** | 0.002 | 0.178 | 0.000 | **0.176** | 0.005 | 1.366 | 0.140 | **0.270** | 0.004 | 0.815 | 0.182 | **0.289** | 0.036 | **0.209** | 0.012 | 0.218 | 0.005 | 1.107 | 0.335 | **0.213** | 0.006 | 0.449 | 0.025 | **0.280** | 0.021 |
| Traffic | 1 | **0.302** | 0.003 | **0.302** | 0.000 | 0.342 | 0.003 | **0.313** | 0.000 | 0.599 | 0.005 | **0.228** | 0.000 | 0.265 | 0.002 | **0.264** | 0.002 | 0.243 | 0.006 | **0.242** | 0.001 | 0.280 | 0.022 | **0.232** | 0.001 | 0.762 | 0.103 | **0.257** | 0.002 |
| | 24 | 0.649 | 0.006 | **0.608** | 0.000 | 0.617 | 0.001 | **0.607** | 0.001 | 0.761 | 0.012 | **0.366** | 0.001 | 0.576 | 0.011 | **0.327** | 0.003 | 0.459 | 0.026 | **0.422** | 0.000 | 0.573 | 0.011 | **0.432** | 0.001 | 1.102 | 0.190 | **0.565** | 0.010 |
| | 48 | 0.769 | 0.001 | **0.681** | 0.000 | 0.707 | 0.001 | **0.680** | 0.000 | 0.824 | 0.019 | **0.403** | 0.004 | 0.683 | 0.008 | **0.369** | 0.003 | 0.517 | 0.011 | **0.457** | 0.001 | 0.621 | 0.052 | **0.454** | 0.001 | 1.001 | 0.126 | **0.604** | 0.017 |
| Weather | 1 | 0.357 | 0.018 | **0.337** | 0.000 | 0.359 | 0.006 | **0.341** | 0.003 | 0.728 | 0.342 | **0.388** | 0.003 | 0.478 | 0.125 | **0.296** | 0.010 | 0.474 | 0.036 | **0.336** | 0.001 | 0.423 | 0.038 | **0.361** | 0.014 | 2.489 | 0.083 | **0.336** | 0.001 |
| | 24 | 1.182 | 0.032 | **1.043** | 0.002 | 1.236 | 0.005 | **1.086** | 0.008 | 1.760 | 0.334 | **1.020** | 0.009 | 1.377 | 0.133 | **0.671** | 0.034 | 1.492 | 0.218 | **1.019** | 0.003 | 1.494 | 0.387 | **0.877** | 0.013 | 2.881 | 0.085 | **1.037** | 0.044 |
| | 48 | 1.702 | 0.082 | **1.447** | 0.002 | 1.634 | 0.012 | **1.434** | 0.002 | 2.620 | 0.474 | **1.415** | 0.018 | 2.699 | 0.334 | **0.909** | 0.017 | 1.740 | 0.065 | **1.435** | 0.003 | 2.033 | 0.423 | **1.308** | 0.007 | 3.039 | 0.029 | **1.419** | 0.025 |

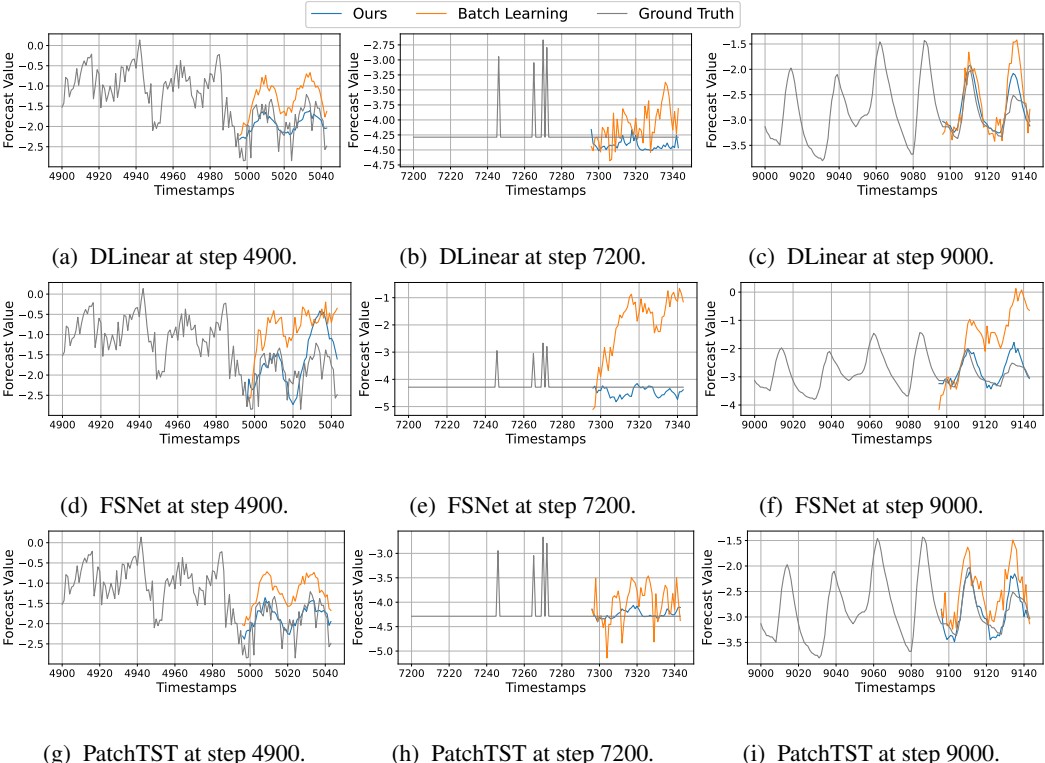

(a) DLinear at step 4900.     (b) DLinear at step 7200.     (c) DLinear at step 9000.

(d) FSNet at step 4900.     (e) FSNet at step 7200.     (f) FSNet at step 9000.

(g) PatchTST at step 4900.     (h) PatchTST at step 7200.     (i) PatchTST at step 9000.

Figure 12: Visualization of the online learning process for ETTh2 at steps 4900, 7200 and 9000.

# E   DETAILED PERFORMANCE COMPARISON RESULTS

## E.1   GENERAL FORECASTING RESULTS

This section provides a detailed performance comparison between our framework and batch learning across various datasets and backbones, as outlined in Section 4.1. While Table 2 highlights average results across all seeds, the standard deviation is also included here in Table 16. The results show performance stability and robustness against variability from random weight initialization, with most standard deviations being less than 0.1.

