# OpenReview forum: "Fast and Slow Streams for Online Time Series Forecasting Without Information Leakage"
_ICLR.cc/2025/Conference — ICLR 2025 Poster_

### Official Review · Reviewer_oqzm · 2024-10-17

**Soundness:** 3
**Presentation:** 3
**Contribution:** 4
**Rating:** 8
**Confidence:** 4

**Summary:**

The paper corrects a fundamental bug in previous papers for Online Time Series Forecasting, proposing a definition of the OTSF task without data leakage. To solve arising challenges of this new scenario, the authors then propose a dual-stream framework (DSOF) for this task, which updates a teacher-student model through residual learning.

In detail, there are two data streams: slow and fast.
- The slow data streams are stored into a buffer in an FIFO basis for updating teacher and student residual model. This requires complete ground truth data for all time steps in data stream. This is called Experience Replay (ER) Update in the paper.
- The fast data stream, designed to utilize data from timestamps where the ground truth for predictions is not yet fully available, updates the student residual model only, according to discord of (the prediction of the teacher-student model in the previous step) and (the ground truth of this step along with the pseudo-label generated by the prediction of the teacher-student model at this step). This is called Temporal Difference (TD) Update in the paper.

**Strengths:**

This work demonstrates several notable strengths.

- Most significantly, the authors point out a significant problem of several works for Online Time Series Forecasting related to information leakage. The bug pointed out here is such fundamental that I did not even believe it existed before I checked the code of several previous works for OTSF and verified the claim of the authors. The redefinition of OTSF problem would definitely make it clearer for the time series community to develop online models that are not only good on benchmarks, but also practical and useful.

In this data-leakage-free setting, there are data points in timestamps whose ground truth for prediction is currently not fully available, which is an arising problem.

- The authors propose the DSOF framework as a logical solution to this emerging problem. This framework will serve as a baseline for future comparisons. The proposed framework provides insights related to differential learning in reinforcement learning, making it more explainable.



- The authors evaluate their proposed framework with extensive experiments conducted across multiple Models (DLinear, FITS, FSNet, OneNet, iTransformer, PatchTST, NSTransformer, etc) and multiple Datasets (ECL, ETT, Traffic, Weather, Exchange, etc) on multiple settings (different forecasting horizons, different learning rates, etc), demonstrating the effectiveness of their method.



The significance of this work is apparent, especially in its correction to the OTSF problem practice, and also in its contribution to point out the arising problem, and propose a method to (partially) deal with this problem.

**Weaknesses:**

A fixed look-back length of 96 may not be entirely convincing. Some models may show a significant advantage at a certain look-back length, but less advantage or even underperformance at longer look-back lengths. It would be better if the authors could follow the practice in the paper of FITS, PatchTST, etc (which are cited by the authors in their paper) to do grid search for the optimal lookback length for each dataset for each forecasting model, and to compare this optimal value against each other.

The authors' approach is understandable, given that (1) some previous work accepted at top conferences for time series forecasting uses similar settings of fixed look-back length and (2) this approach would require much larger computation resource. However, I believe it would benefit the time series community if new papers utilized grid search and comparing with baselines across different look-back lengths, though this would not affect the contribution and significance of this work.

**Questions:**

- Could you conduct a grid search for different look-back horizons on selected datasets, models, and forecasting horizons to demonstrate robustness with respect to look-back horizon? I'm not suggesting that you repeat all experiments. Conducting grid searches for a subset, particularly with smaller models and datasets (e.g., DLinear, iTransformers on ETTh1, Exchange; H=24), would be computationally feasible and enhance the robustness of your results.

- I noted the experiments on different buffer-size in appendix. Would it be possible that these experiments could partially represent or replace experiments about different look-back horizon?

- Recent work has proposed Foundation Models for Time Series Forecasting, and there are papers discussing Test Time Adaptation for Time Series Forecasting. Would it be possible that the predictions of base models can be improved by the DSOF framework?

---

> ### Comment · Reviewer_oqzm · 2024-11-13
> **Further advice on clarification and illustration about the data leakage problem and its importance**
>
> I think **the problem about data leakage is very important in practice**. **I further suggest the authors provide some concrete examples (maybe in the appendix) to better illustrate this problem, and emphasize its importance**. For example:
>
> In practice, say I have a model to predict the temperature of the next 24 days, and I want to update my model everyday. Yesterday my model has a prediction of today and the next 23 days, according to which I would like to update my model today. Of-course only today's ground truth is available: meanwhile in previous work they update model parameter **today** according to gt value of **today and the next 23 days**, and that's **data leakage** described in this work. Unfortunately, this problem of data leakage has been found in previous work for OTSF.
>
> **It is an important problem not only in evaluating models, but also in practice**. Take quantitative trading as an example, evaluating a prediction model with such data leakage (as in previous work for OTSF) in backtesting would lead to over estimation. In trading firms they would avoid such data leakage to avoid fake profits in backtesting, but many of previous OTSF researches are not avoiding this issue, **making previous research results not that practical in real scenarios**.
>
> By pointing out this problem and provide corresponding methods for OTSF, **this work can potentially close the gap between OTSF researches and real scenarios for OTSF**.
>
> The description of the data leakage problem in this work is indeed complicated, and the importance of this problem might not be completely emphasized in this work. Therefore, **I suggest the authors provide some concrete examples and emphasize the importance of this problem in practice**.

---

> > ### Author Response · Authors · 2024-11-22
> >
> > Thank you for your suggestion! The examples you provided are excellent and valuable. We have revised the introduction to reduce the use of mathematical notations [*Section 1.1, line 041*] and have updated the figure [*Section 1.1, line 062*] to better illustrate the information leak problem, making it more accessible to readers from different fields. Additionally, we have relocated the formulation of the new OTSF setting to the methodology section [*Section 3.1, line 191*].

---

> ### Author Response · Authors · 2024-11-22
> **Response to Reviewer oqzm**
>
> ### Q1: Different look-back window lengths
> > Could you conduct a grid search for different look-back horizons on selected datasets, models, and forecasting horizons to demonstrate robustness with respect to look-back horizon? I'm not suggesting that you repeat all experiments. Conducting grid searches for a subset, particularly with smaller models and datasets (e.g., DLinear, iTransformers on ETTh1, Exchange; H=24), would be computationally feasible and enhance the robustness of your results.
>
> As included in [*Section A.8, line 1173*],
>
> We conduct a grid search over the look-back window length $L \in {24, 48, 96, 192}$. Contrary to the common belief that longer lookback windows provide more information and thus enhance forecasting performance, our findings suggest otherwise. The results reveal that in most cases, model performance declines as the look-back window length increases. Interestingly, a look-back window length of 24 often results in better performance, regardless of the horizon window.
>
> Impact of lookback window length $L \in \{24, 48, 96, 192\}$:
> |||  DLinear  || | | iTransformer | | || PatchTST  | | ||
> | -------- | ----- | :-------: | ----- | --------- | --------- | :----------: | --------- | --------- | ----- | :-------: | --------- | --------- | ----- |
> | Datasets | H$/$L | 24| 48 | 96 | 192| 24 | 48 | 96 | 192| 24| 48 | 96 | 192|
> | ECL | 1|2.802| 2.258 | 2.065| **2.010** | 2.341| **2.282** | 2.430| 2.349 |2.603| 2.265| **2.244** | 2.476 |
> || 24 |5.396| 4.845 | **4.737** | 4.980| 5.651| **5.101** | 5.155| 5.571 |5.247| **5.128** | 5.169| 5.516 |
> || 48 |6.563| 6.270 | **6.181** | 6.778| 6.736| 6.324| **6.015** | 7.734 | **6.333** | 6.366| 6.665| 8.461 |
> | ETTh2 | 1| **0.364** | 0.369 | 0.365| 0.371|  **0.371**| 0.398| 0.384| 0.374 |0.385| 0.388| **0.382** | 0.394 |
> || 24 | **1.664** | 1.666 | 1.701| 1.891|  **1.658**| 1.704| 1.869| 2.103 |1.859| **1.850** | 1.925| 2.135 |
> || 48 | **2.839** | 2.951 | 3.082| 3.552|  **2.801**| 2.972| 3.465| 4.035 |3.506| **3.244** | 4.473| 5.259 |
> | Ex. | 1| **0.009** | 0.009 | 0.009| 0.010|  **0.009**| 0.011| 0.010| 0.014 |0.010| **0.009** | 0.011| 0.010 |
> || 24 | **0.092** | 0.093 | 0.095| 0.101|  **0.095**| 0.100| 0.110| 0.115 | **0.093** | 0.101| 0.103| 0.107 |
> || 48 | **0.184** | 0.186 | 0.192| 0.205|  **0.206**| 0.228| 0.218| 0.305 | **0.188** | 0.202| 0.213| 0.242 |
>
> ### Q2: Relationship of buffer size and look-back window lengths
> > I noted the experiments on different buffer-size in appendix. Would it be possible that these experiments could partially represent or replace experiments about different look-back horizon?
>
> Intuitively, buffer sizes and look-back window size should exhibit some correlation. If a larger buffer size improves MSE performance, it implies that historical data streams are related to the current distribution being predicted. Similarly, if a longer look-back window enhances performance, it indicates that incorporating more past information is beneficial for prediction.
>
> However, while larger buffer sizes consistently yield better results [*Section A.5.1, line 943*], a longer look-back window does not always lead to improved outcomes, as shown in the experiments in [*Section A.8, line 1173*]. This discrepancy might be due to the ER strategy used. Recall that the slow stream must wait for $L+H$ steps to ensure that only data pairs with all ground truth values are included. A larger look-back window $L$ means the data sequences have to wait longer before being added to the buffer, potentially resulting in a less informative slow stream and reducing the effectiveness of the ER procedure. Consequently, in this scenario, experiments on buffer size cannot directly relate with the results of look-back window sizes.
>
> ### Q3: Adapting DSOF to foundation models
> > Recent work has proposed Foundation Models for Time Series Forecasting, and there are papers discussing Test Time Adaptation for Time Series Forecasting. Would it be possible that the predictions of base models can be improved by the DSOF framework?
>
> Foundation models for time series forecasting typically involve two phases: pre-training on a large dataset and adaptation to specific tasks or datasets. DSOF could potentially enhance the adaptation phase of these models. By enabling online adaptation, the DSOF framework can be particularly beneficial for datasets where the temporal domain of the time series changes rapidly over time.
>
> Combining the advantages of foundation models and online learning has the potential to create more accurate forecasts that can be applied effectively in real-life situations. During the pre-training phase, foundation models develop strong generalization capabilities. DSOF can further ensure that the model remains accurate and relevant by adapting in real-time to new data. However, this is a broad area that warrants further exploration and research to fully understand its potential impact.

---

> > ### Comment · Reviewer_oqzm · 2024-11-23
> >
> > I would like to thank the authors for their rebuttals. I think they address my concerns.
> >
> > (Moreover, I think it is not weird that longer lookback window size may provide worse result. Even for the most classic time series forecasting tasks, there does exist an optimal lookback horizon for time series forecasting dataset of a certain size, and it increases with available amount of training data: the underlying mechanism is similar to overfitting. This is why tricks like patches or low-pass-filters used in PatchTST or FITS improve model performance.)

---

### Official Review · Reviewer_7cqR · 2024-10-21

**Soundness:** 3
**Presentation:** 2
**Contribution:** 3
**Rating:** 6
**Confidence:** 4

**Summary:**

The paper addresses challenges in online time series forecasting (OTSF), particularly the issue of information leakage, where models use historical data for both prediction and evaluation, leading to biased performance assessments. The authors redefine OTSF to focus exclusively on predicting future unknown time steps without using backpropagated data for evaluation, aligning the framework more closely with real-world forecasting tasks.

**Strengths:**

The paper presents a novel redefinition of online time series forecasting (OTSF), specifically addressing the issue of information leakage, which has not been adequately tackled in prior research. By focusing on predicting future time steps without evaluating on previously backpropagated data, the authors set a new standard for OTSF, ensuring a more realistic and reliable forecasting setting. The introduction of the dual-stream framework, DSOF, which combines experience replay (ER) and temporal difference (TD) learning, is an innovative approach that allows the model to adapt effectively to temporal shifts and generalize better to real-world scenarios.


The study is well-executed, with thorough experimentation that covers multiple datasets and a variety of model architectures. The results clearly demonstrate the advantages of the proposed DSOF framework over traditional batch learning methods, showing significant improvements across diverse real-world scenarios. The experiments also include ablation studies, highlighting the importance of core components such as ER and TD loss, thus providing a detailed understanding of how different design choices contribute to the model’s performance. The methodological framework is backed by rigorous empirical evidence, lending credibility to the findings.

**Weaknesses:**

The results of the baselines, especially the comparison between OneNet and FSNet, are unexpected and raise concerns about the experimental setup. Specifically, OneNet, which is theoretically more robust and effective, shows worse performance compared to FSNet, especially on datasets like ECL, which have many variables. This discrepancy is puzzling because OneNet should typically outperform FSNet due to its more comprehensive handling of cross-variable and cross-time dependencies. The authors should clarify the hyperparameter settings used for baseline models and ensure that they have been adequately optimized for each baseline. A detailed explanation of why OneNet underperforms would be valuable, as it could indicate either a misconfiguration or potential issues in the experimental protocol. Additionally, including a sensitivity analysis for hyperparameters could help identify whether the results are consistent or if specific settings unduly favor certain models


While the paper argues that existing OTSF settings introduce information leakage, it overlooks that many practical use cases update the model immediately with each new data point rather than accumulating a sequence. Although the authors highlight the risks of information leakage, real-world scenarios often prioritize fast adaptation to incoming data, even if it means some overlap in training and evaluation periods. This practice, while not perfect, remains a common approach because it ensures the model quickly adapts to recent changes. The authors should address why the proposed method of avoiding information leakage offers a practical advantage, despite differing from common real-world practices. It would also help to show more use cases where avoiding this overlap leads to clear benefits, thus making a stronger case for why users should adopt the new setting.

The paper does not provide a thorough analysis of the computational cost associated with running DSOF in real-time environments. Given the dual-stream approach and reliance on replay buffers, the method could potentially introduce higher computational demands, especially when dealing with large datasets or long lookback windows. Include a detailed assessment of the computational overhead introduced by the dual-stream approach. Specifically, measure the latency, memory usage, and training time, comparing them with baseline batch learning and other online learning frameworks. This would provide a clearer understanding of the trade-offs involved and help identify scenarios where DSOF is more efficient or where further optimizations are needed.

In Figure 3, the combined use of ER and TD loss leads to a significant performance boost, but the individual contributions of each are less apparent. This pattern is confusing because it suggests that the benefits of each component alone are minimal, raising questions about their necessity if they do not independently improve performance. Without a clearer understanding of how these components interact, it is difficult to assess their true value within the framework.   The authors should provide a more detailed analysis explaining why ER and TD loss, when used together, show such a marked improvement over their separate contributions. This might include examining how these methods complement each other and identifying specific scenarios where one is more effective than the other. Additional ablation studies or visualizations showing how each component contributes to different aspects of the model’s learning could offer deeper insights into their interactions.

**Questions:**

.see weakness

---

> ### Author Response · Authors · 2024-11-22
> **Response to Reviewer 7cqR (Part 1)**
>
> ### Q1: Concerns Regarding OneNet's Performance
>
> > The results of the baselines, especially the comparison between OneNet and FSNet, are unexpected and raise concerns about the experimental setup. Specifically, OneNet, which is theoretically more robust and effective, shows worse performance compared to FSNet, especially on datasets like ECL, which have many variables. This discrepancy is puzzling because OneNet should typically outperform FSNet due to its more comprehensive handling of cross-variable and cross-time dependencies.
>
> After carefully reviewing the code and comparing it with the version in the official repository, we discovered that the OCP block of OneNet was not trained as intended. After making the necessary corrections, we observed an improvement in the results, which are now more reasonable. (This also confirms the effectiveness of the OCP block in OneNet.) In most cases, there was a significant reduction in MSE. We appreciate for bringing this to our attention.
>
> The results have been updated in the main text, which includes the average of 5 repeated runs of the experiment [*Section 4.1, line 378*], as well as in the appendix, which displays the standard deviation [*Section E.1, line 1289*].
>
> |          |    |     FSNet    |        |       |       |    OneNet  (Before retraining)  |        |       |       |    OneNet (After retraining)    |        |       |       |
> |----------|----|:------------:|--------|-------|-------|:------------:|--------|-------|-------|:------------:|--------|-------|-------|
> |          | H  | Batch Learn. | SD     | DSOF  | SD    | Batch Learn. | SD     | DSOF  | SD    | Batch Learn. | SD     | DSOF | SD    |
> | ECL      | 1  | 3.8e+2       | 1.5e+1 | 2.330 | 0.056 | 3.3e+2       | 2.7e+2 | 2.411 | 0.148 | 3.2e+1       | 1.6e+1 | 4.733 | 0.746 |
> | ECL      | 24 | 4.7e+2       | 2.816  | 5.475 | 0.043 | 1.7e+2       | 2.0e+1 | 5.523 | 0.164 | 8.5e+1       | 8.561  | 4.510 | 0.157 |
> | ECL      | 48 | 4.8e+2       | 3.927  | 7.000 | 0.133 | 1.9e+2       | 3.3e+1 | 7.347 | 0.189 | 1.5e+2       | 1.8e+1 | 5.943 | 0.090 |
> | ETTh2    | 1  | 1.1e+1       | 1.772  | 0.431 | 0.012 | 2.021        | 0.826  | 0.581 | 0.039 | 2.531        | 0.166  | 0.548 | 0.048 |
> | ETTh2    | 24 | 1.9e+1       | 1.693  | 3.189 | 0.090 | 6.156        | 0.767  | 3.018 | 0.038 | 7.017        | 0.966  | 2.363 | 0.087 |
> | ETTh2    | 48 | 2.3e+1       | 2.381  | 5.754 | 0.271 | 9.668        | 2.307  | 5.484 | 0.361 | 9.790        | 1.577  | 4.037 | 0.245 |
> | ETTm1    | 1  | 0.190        | 0.019  | 0.121 | 0.008 | 0.148        | 0.009  | 0.132 | 0.002 | 0.156        | 0.019  | 0.096 | 0.008 |
> | ETTm1    | 24 | 1.500        | 0.163  | 0.609 | 0.011 | 0.977        | 0.073  | 0.603 | 0.009 | 1.094        | 0.056  | 0.418 | 0.035 |
> | ETTm1    | 48 | 2.279        | 0.268  | 0.843 | 0.017 | 1.437        | 0.071  | 0.802 | 0.031 | 1.588        | 0.120  | 0.554 | 0.024 |
> | Exchange | 1  | 0.034        | 0.008  | 0.010 | 0.000 | 0.026        | 0.007  | 0.011 | 0.000 | 0.024        | 0.004  | 0.009 | 0.000 |
> | Exchange | 24 | 0.754        | 0.230  | 0.120 | 0.002 | 0.698        | 0.237  | 0.219 | 0.008 | 0.487        | 0.079  | 0.152 | 0.009 |
> | Exchange | 48 | 1.366        | 0.140  | 0.270 | 0.004 | 1.027        | 0.282  | 0.469 | 0.047 | 0.815        | 0.182  | 0.289 | 0.036 |
> | Traffic  | 1  | 0.599        | 0.005  | 0.228 | 0.000 | 0.316        | 0.005  | 0.217 | 0.001 | 0.265        | 0.002  | 0.264 | 0.002 |
> | Traffic  | 24 | 0.761        | 0.012  | 0.366 | 0.001 | 0.580        | 0.005  | 0.367 | 0.002 | 0.576        | 0.011  | 0.327 | 0.003 |
> | Traffic  | 48 | 0.824        | 0.019  | 0.403 | 0.004 | 0.682        | 0.011  | 0.398 | 0.002 | 0.683        | 0.008  | 0.369 | 0.003 |
> | Weather  | 1  | 0.728        | 0.342  | 0.388 | 0.003 | 0.379        | 0.016  | 0.384 | 0.066 | 0.478        | 0.125  | 0.296 | 0.010 |
> | Weather  | 24 | 1.760        | 0.334  | 1.020 | 0.009 | 1.534        | 0.285  | 0.949 | 0.027 | 1.377        | 0.133  | 0.671 | 0.034 |
> | Weather  | 48 | 2.620        | 0.474  | 1.415 | 0.018 | 2.322        | 0.237  | 1.300 | 0.021 | 2.699        | 0.334  | 0.909 | 0.017 |

---

> > ### Comment · Reviewer_7cqR · 2024-11-26
> > **response to the author**
> >
> > Some results are still a bit puzzling. For example, on ETTh2, why does the OCP block perform better without training? A similar phenomenon also occurs on Weather. Additionally, under the DSOF setting, why is the performance of OneNet significantly worse than "Before retraining"?

---

> ### Author Response · Authors · 2024-11-22
> **Response to Reviewer 7cqR (Part 2)**
>
> ### Q2: The Neccessity of Redefining the OTSF setting
>
> > While the paper argues that existing OTSF settings introduce information leakage, it overlooks that many practical use cases update the model immediately with each new data point rather than accumulating a sequence. Although the authors highlight the risks of information leakage, real-world scenarios often prioritize fast adaptation to incoming data, even if it means some overlap in training and evaluation periods. This practice, while not perfect, remains a common approach because it ensures the model quickly adapts to recent changes. The authors should address why the proposed method of avoiding information leakage offers a practical advantage, despite differing from common real-world practices. It would also help to show more use cases where avoiding this overlap leads to clear benefits, thus ...
>
> Before addressing the core questions regarding the necessity of our proposed setting, we would like to ensure clarity by providing a more detailed description of our approach. Let us assume the input and output windows have lengths $L = 5$ and $H = 4$, respectively.
>
> - **Previous Setting**: At time $t=10$, the model uses data from $t=3$ to $t=7$ to predict values from $t=8$ to $t=11$. Once the actual data for $t=11$ is available, predictions made at $t=10$ are evaluated against the ground truth, and the model is updated using the data from $t=8$ to $t=11$. This process repeats at $t=12$, advancing predictions by one time unit.
> - **Our Setting**: At $t=10$, the model uses data from $t=6$ to $t=10$ to forecast values from $t=11$ to $t=14$. At $t=11$, the model is updated with ground truths only up to $t=11$. The predictions made at $t=10$ are evaluated on or after $t=14$, when all actual data from $t=11$ to $t=14$ is available. This process repeats at $t=12$, advancing predictions by one time unit.
>
> You can refer to the updated problem statement of OTSF setting in [*Section 1.1, line 041*] and the revised figure at [*Section 1.1, line 062*] for further clarification.
>
> Clarifications:
> 1. "While the paper argues that existing OTSF settings introduce information leakage, it overlooks that many practical use cases update the model immediately with each new data point rather than accumulating a sequence."
>
>     Similar to FSNet and OneNet, DSOF also makes predictions in a moving window of 1 and updates the model immediately when each new data point arrives. For instance, at $t=10$, we predict $t=11, 12, 13, 14$. When the ground truth at $t=11$ becomes available, DSOF uses this data to update the model parameters, aiding in the prediction of the next sequence $t=12, 13, 14, 15$. At $t=11$, even if we do not have actual data of $t = 12, 13,14$, the model can still be updated immediately by the fast data stream, which involves creating psuedo labels that concatenates the ground truth at $t = 11$ and the teacher model's predictions for $t= 12, 13, 14$
>
> 2. "why the proposed method of avoiding information leakage offers a practical advantage"
>
>     Avoiding information leakage in model evaluation is crucial for ensuring accurate and reliable performance assessments in real-world applications.
>
>     In the previous setting, information leakage occurs because the evaluation at $t=12$ includes data points from $t=9$ to $t=11$, which have already been used to optimize the model parameters at $t=11$. This overlap in time steps results in biased evaluation outcomes, leading to an overestimation of the model's effectiveness in real-world applications.
>
>     For instance, as highlighted by reviewer oqzm, in quantitative trading, evaluating a prediction model with data leakage can lead to overestimation during backtesting. Trading firms avoid such leakage to prevent false profits in backtesting, underscoring the impracticality of previous OTSF research results in real scenarios.
>
> 3. "show more use cases where avoiding this overlap leads to clear benefits"
>
>     Our redefined setting provides additional advantages by allowing for $H$ future predictions. In contrast, previous settings, even with $H > 1$, only forecast one future step at any given timestamp. For instance, at $t=10$, the model predicts values from $t=8$ to $t=11$, even though the ground truth for $t=8$ to $t=10$ is already known. Consequently, setting $H > 1$ in the old setting is effectively the same as setting $H = 1$. Setting $H > 1$ introduces unnecessary computational costs.
>
>     It is important to note that our setting also involves overlap in predictions, as input data sequences are generated using a moving window with a step size of 1. However, the key distinction is that the previous setting overlaps in timestamps with known ground truth, whereas our setting overlaps in timestamps without ground truth. This means that when $H > 1$ is set, the framework predicts the data point at $t=10$ multiple times before the time actually reaches $t=10$.

---

> > ### Comment · Reviewer_7cqR · 2024-11-26
> > **response to the author**
> >
> > The author answered this question very well.

---

> ### Author Response · Authors · 2024-11-22
> **Response to Reviewer 7cqR (Part 3)**
>
> ### Q3: Computational costs of DSOF
> > The paper does not provide a thorough analysis of the computational cost associated with running DSOF in real-time environments. Given the dual-stream approach and reliance on replay buffers, the method could potentially introduce higher computational demands, especially when dealing with large datasets or long lookback windows. Include a detailed assessment of the computational overhead introduced by the dual-stream approach. Specifically, measure the latency, memory usage, and training time, comparing them with baseline batch learning and other online learning frameworks. This would provide a clearer understanding of the trade-offs involved and help identify scenarios where DSOF is more efficient or where further optimizations are needed.
>
> In addition to the online training framework of FSNet and OneNet (which we refer to as DGrad), we also investigated other online learning frameworks, specifically TFCL and DER++. Unlike DSOF, these methods — DGrad, TFCL, and DER++ — do not incorporate a student model. To facilitate a fair comparison, we also present an alternative version of DSOF that excludes the student model, referred to as DSOF w/o $\theta^{(S)}$. More details can be found in [*Section 4.2, line 411*], and a comparison of MSE performance is presented in [*Section 4.2, line 486*].
>
> Then it is followed by an evaluation of runtime in **milliseconds per iteration (ms / itr)** in [*Section A.1.1, line 686*], GPU memory usage in **mebibyte (MiB)** in [*Section A.1.2, line 701*].
>
> Wall time comparison (ms/itr) across various frameworks:
> | |  ECL  | | ETTh2 | | Traffic ||
> | ------------------------- | :---: | :--: | :---: | :--: | :-----: | :---: |
> | Framework   | Mean  |  SD  | Mean  |  SD  |  Mean   |  SD   |
> | DGrad| 5.58  | 0.18 | 4.63  | 0.14 |  11.03  | 1.06  |
> | DER++| 13.79 | 1.08 | 12.71 | 3.50 |  23.82  | 3.37  |
> | TFCL | 23.28 | 1.69 | 13.02 | 0.33 |  30.89  | 2.76  |
> | DSOF (w/o $\theta^{(S)}$) | 23.16 | 1.40 | 8.13  | 0.12 |  51.66  | 29.09 |
> | DSOF | 30.76 | 1.06 | 12.04 | 0.16 |  63.96  | 12.42 |
>
> Processor time (ms/itr) comparison across various frameworks:
> | |  ECL   || ETTh2  || Traffic ||
> | ------------------------- | :----: | :---: | :----: | :---: | :-----: | :---: |
> | Framework   |  Mean  |  SD   |  Mean  |  SD   |  Mean   |  SD   |
> | DGrad|  4.90  | 0.36  |  3.67  | 0.18  | 166.87  | 19.09 |
> | DER++| 193.29 | 17.46 | 155.89 | 26.68 | 373.74  | 30.84 |
> | TFCL | 337.87 | 17.38 | 137.98 | 3.51  | 551.64  | 79.99 |
> | DSOF (w/o $\theta^{(S)}$) | 358.31 | 18.71 |  7.32  | 0.32  | 596.36  | 58.51 |
> | DSOF | 478.45 | 31.96 | 10.91  | 1.20  | 800.70  | 10.74 |
>
> Comparison of GPU memory usage (MiB) during the online phase across different online learning frameworks:
> | Framework   | ECL | ETTh2 | Traffic |
> | ------------------------- | :-: | :---: | :-----: |
> | DGrad| 386 |  346  |   436   |
> | DER++| 424 |  346  |   542   |
> | TFCL | 392 |  346  |   440   |
> | DSOF (w/o $\theta^{(S)}$) | 386 |  346  |   436   |
> | DSOF | 400 |  348  |   508   |
>
>
> We recognize that our method has certain drawbacks in terms of runtime. To address this, we conducted an analysis of how the main components of our method contribute to runtime, as presented in [*Section A.2.4, line 794*].
>
> Wall time comparison (ms/itr) for the two main components of the framework:
> | | |||  ECL  | | ETTh2 | | Traffic ||
> | :-----------: | :-----------: | :-: | :-: | :---: | :--: | :---: | :--: | :-----: | :---: |
> | Teacher Model | Student Model | ER  | TD  | Mean  |  SD  | Mean  |  SD  |  Mean   |  SD   |
> |    DLinear    | MLP |  X  |  X  | 5.58  | 0.18 | 4.63  | 0.14 |  11.03  | 1.06  |
> |    DLinear    | MLP |  V  |  X  | 27.98 | 2.18 | 8.12  | 0.10 |  43.74  | 3.45  |
> |    DLinear    | MLP |  X  |  V  | 8.32  | 0.51 | 6.53  | 0.44 |  17.85  | 1.60  |
> |    DLinear    | MLP |  V  |  V  | 30.76 | 1.06 | 12.04 | 0.16 |  63.96  | 12.42 |
>
> Processor time comparison (ms/itr) for the two main components of the framework:
> | | |||  ECL   || ETTh2 | | Traffic ||
> | :-----------: | :-----------: | :-: | :-: | :----: | :---: | :---: | :--: | :-----: | :---: |
> | Teacher Model | Student Model | ER  | TD  |  Mean  |  SD   | Mean  |  SD  |  Mean   |  SD   |
> |    DLinear    | MLP |  X  |  X  |  4.90  | 0.36  | 3.67  | 0.18 | 166.87  | 19.09 |
> |    DLinear    | MLP |  V  |  X  | 434.26 | 16.37 | 7.58  | 0.68 | 637.02  | 19.47 |
> |    DLinear    | MLP |  X  |  V  |  7.00  | 0.22  | 5.30  | 0.05 | 289.61  | 27.26 |
> |    DLinear    | MLP |  V  |  V  | 478.45 | 31.96 | 10.91 | 1.20 | 800.70  | 10.74 |
>
> The findings clearly indicate that ER has a greater impact on runtime than TD loss. One potential strategy to mitigate this is to use a smaller replay batch size. Our experiments, outlined in [*Section A.5.2, line 969*], demonstrate that larger batch sizes do not yield significant performance improvements. Therefore, employing a smaller batch size of 8 or 16 in ER can help balance efficiency and accuracy.

---

> > ### Comment · Reviewer_7cqR · 2024-11-26
> > **response to the author**
> >
> > Because computational overhead is indeed a big problem in online forecasting, this article does have a large computational overhead. However, considering that the author provides future improvement plans and the method has achieved good results, this problem should not be considered a major flaw of this article.

---

> ### Author Response · Authors · 2024-11-22
> **Response to Reviewer 7cqR (Part 4)**
>
> ### Q4: Interaction of ER and TD Loss
>
> > In Figure 3, the combined use of ER and TD loss leads to a significant performance boost, but the individual contributions of each are less apparent. This pattern is confusing because it suggests that the benefits of each component alone are minimal, raising questions about their necessity if they do not independently improve performance. Without a clearer understanding of how these components interact, it is difficult to assess their true value within the framework. The authors should provide a more detailed analysis explaining why ER and TD loss, when used together, show such a marked improvement over their separate contributions. This might include examining how these methods complement each other and identifying specific scenarios where one is more effective than the other. Additional ablation studies or visualizations showing how each component contributes to different aspects of the model’s learning could offer deeper insights into their interactions.
>
> In [*Section A.2.3, line 772*], we explore how the methods complement each other and identify specific scenarios where one is more effective than the other. We provide visualizations to illustrate these interactions.
>
> For sequences with regular cycles, ER helps reduce variance by providing stable and confident predictions through training with batches of historical sequences. On the other hand, the TD loss component relies on pseudo labels for updates, where only the initial data point is based on ground truth, and the remaining points are less certain predictions from the teacher model. Relying solely on TD loss can lead to significant variance in predictions, even for predictable sequences with regular cycles.
>
> For immediate changes in data distribution, TD loss can make rapid adaptations. However, the ER component, which depends on historical data, may make outdated assumptions about the current distribution, potentially leading to confident but inaccurate predictions in rapidly changing environments.
>
> The interaction between ER and TD loss resembles a bias-variance trade-off. By utilizing both components, the model can harness their strengths to balance this trade-off, resulting in more accurate and reliable predictions across different data distributions.

---

> ### Author Response · Authors · 2024-11-26
> **Response to Reviewer 7cqR (Part 2 - Update)**
>
> We believe that the previous table in our earlier response was unclear due to the excessive number of columns. We have rewritten the table by removing the sub-columns for (1) FSNet and (2) OneNet trained under the batch learning setting, as well as the (3) standard deviation (SD) columns. The revised table clearly demonstrates that OneNet performs better with OCP training. Under the DSOF setting, the performance of OneNet shows significant improvement after retraining. Only one or two instances of prediction length 1 are underperforming, which may be attributed to hyperparameter issues.
>
> OneNet's performance with **DSOF**:
> |  |  | OneNet (Before retraining) | OneNet (After retraining) |
> |---|---|---|---|
> |  | H | **DSOF** | **DSOF** |
> | ECL | 1 | **2.411** | 4.733 |
> |  | 24 | 5.523 | **4.510** |
> |  | 48 | 7.347 | **5.943** |
> | ETTh2 | 1 | 0.581 | **0.548** |
> |  | 24 | 3.018 | **2.363** |
> |  | 48 | 5.484 | **4.037** |
> | ETTm1 | 1 | 0.132 | **0.096** |
> |  | 24 | 0.603 | **0.418** |
> |  | 48 | 0.802 | **0.554** |
> | Exchange | 1 | 0.011 | **0.009** |
> |  | 24 | 0.219 | **0.152** |
> |  | 48 | 0.469 | **0.289** |
> | Traffic | 1 | **0.217** | 0.264 |
> |  | 24 | 0.367 | **0.327** |
> |  | 48 | 0.398 | **0.369** |
> | Weather | 1 | 0.384 | **0.296** |
> |  | 24 | 0.949 | **0.671** |
> |  | 48 | 1.300 | **0.909** |
>
> One possible reason it might have been observed that "before retraining" performs better than "after retraining" could be that the "Batch Learn" column was looked into instead of the "DSOF" subcolumn in the previous table. Given that the OneNet's analysis is based on regret minimization in online learning, the OCP block may not be designed for batch learning. This is similar to FSNet, which tends to perform significantly better in online scenarios compared to the batch learning setting.
>
> OneNet's performance in **Batch Learning**:
>
> |  |  | OneNet (Before retraining) | OneNet (After retraining) |
> |---|---|---|---|
> |  | H | **Batch Learn.** | **Batch Learn.** |
> | ECL | 1 | 3.3e+2 | **3.2e+1** |
> | ECL | 24 | 1.7e+2 | **8.5e+1** |
> | ECL | 48 | 1.9e+2 | **1.5e+2** |
> | ETTh2 | 1 | **2.021** | 2.531 |
> | ETTh2 | 24 | **6.156** | 7.017 |
> | ETTh2 | 48 | **9.668** | 9.790 |
> | ETTm1 | 1 | **0.148** | 0.156 |
> | ETTm1 | 24 | **0.977** | 1.094 |
> | ETTm1 | 48 | **1.437** | 1.588 |
> | Exchange | 1 | 0.026 | **0.024** |
> | Exchange | 24 | 0.698 | **0.487** |
> | Exchange | 48 | 1.027 | **0.815** |
> | Traffic | 1 | 0.316 | **0.265** |
> | Traffic | 24 | 0.580 | **0.576** |
> | Traffic | 48 | **0.682** | 0.683 |
> | Weather | 1 | **0.379** | 0.478 |
> | Weather | 24 | 1.534 | **1.377** |
> | Weather | 48 | **2.322** | 2.699 |

---

> > ### Comment · Reviewer_7cqR · 2024-11-27
> > **respond to the author**
> >
> > The baseline implementation and performance seem to be fine, and my biggest concern has been resolved, so I am inclined to accept it.

---

### Official Review · Reviewer_Qe4w · 2024-11-01

**Soundness:** 2
**Presentation:** 1
**Contribution:** 2
**Rating:** 6
**Confidence:** 2

**Summary:**

This paper discusses a dual-stream framework for online forecasting (DSOF). Two modules separate the learning process into two streams: a “slow stream” which updates with complete data, and a “fast stream” which adapts to the most recent data. Using a teacher-student framework， the proposed model aims to improve forecasting performance by balancing adaptation to new data with robustness against noise. Numerical experiments are provided.

**Strengths:**

This paper tries to address an important topic in time series forecasting via online model adjusting to hedge the distribution drifting issue.

**Weaknesses:**

1. The presentation of this paper could be improved. According to the current draft, the novelty appears limited. The way to utilize teacher-student style models and experience replay seems quite standard. I suggest authors include some toy examples or a theoretical analysis to motivate the necessity proposed method from a more principled way.

2. Sample code is not provided to assist the reviewer in verifying reproducibility.

3. Can authors elaborate more on how the proposed model addresses the information leakage? It seems not being well discussed in methodology section (section 3).

**Questions:**

Please see the weakness part.

At the current stage, the novelty is insufficient for acceptance at a top-tier machine learning conference, and I tend to recommend rejection. However, I am open to reevaluating this work based further discussion.

---

> ### Author Response · Authors · 2024-11-22
> **Response to Reviewer Qe4w (Part 1)**
>
> ### Overview of the two novelties
>
> > According to the current draft, the novelty appears limited.
>
> We appreciate your feedback and would like to clarify the novelties in our work. Our research presents two key contributions that address the shortcomings of existing methods.
>
> First, we redefined the OTSF setting to prevent information leakage. In the redefined setting, data included for evaluation cannot be used for model parameter updates initially. This ensures that the evaluation process remains unbiased.
>
> Second, based on this new OTSF setting, we introduce DSOF, a model-agnostic framework that operates within this redefined setting. It allows for learning the most up-to-date information without requiring all ground truth data points of the sequence to be available beforehand.

---

> ### Author Response · Authors · 2024-11-22
> **Response to Reviewer Qe4w (Part 2)**
>
> ### Novelty 1: redefined the OTSF setting to close gap of research and real-life appliations
>
> > how the proposed model addresses the information leakage?
>
> **Previous OTSF settings** used in existing OTSF works ( such as FSNet and OneNet):
> - At time $t=10$, the model uses data from $t=3$ to $t=7$ to predict values from $t=8$ to $t=11$. Once the actual data for $t=11$ is available, predictions made at $t=10$ are evaluated against the ground truth, and the model is updated using the data from $t=8$ to $t=11$. This process repeats at $t=12$, advancing predictions by one time unit.
>
> **Two problems** can be identified from using the previous setting:
> 1. Information leakage occurs because the evaluation at $t=12$ includes data points from $t=9$ to $t=11$, which have already been used for parameter optimization at $t=11$. This overlap leads to biased evaluation results and overestimates the model's effectiveness in real-world scenarios.
>
> 1. The model forecasts only one future step at any given timestamp. For instance, at $t=10$, it predicts values from $t=8$ to $t=11$, even though the ground truth from $t=8$ to $t=10$ is already known. Consequently, setting $H > 1$ in the old setting is effectively the same as setting $H = 1$. Setting $H > 1$ introduces redundant computational costs.
>
> To solve the above problems, we **redefined the setting**:
> At $t=10$, the model uses data from $t=6$ to $t=10$ to forecast values from $t=11$ to $t=14$. At $t=11$, the model can only be updated with ground truths up to $t=11$. The predictions made at $t=10$ are evaluated on or after $t=14$, when all actual data from $t=11$ to $t=14$ is available. This process repeats at $t=12$, advancing predictions by one time unit.
>
> For further details, please refer to the updated description of the OTSF leakage problem in [*Section 1.1, line 041*] and the illustration in [*Section 1.1, line 061*]. The description of the new setting can be found in [*Section 1.1, line 086*].
>
>
> ### Novelty 2:  applying DSOF to the modified OTSF setting
>
> > The way to utilize teacher-student style models and experience replay seems quite standard. I suggest authors include some toy examples or a theoretical analysis to motivate the necessity proposed method from a more principled way.
>
>
> Our primary contribution is redefining the OTSF setting to eliminate information leakage. Then we realized that this setting poses challenges because the predicted sequence cannot be fully updated in a supervised manner due to the lack of ground truth for subsequent time steps, as discussed in [*Section 1.2, line 095*]. For example, at $t=10$, we predict $t=11, 12, 13, 14$. At $t = 11$, when the ground truth at $t = 11$ becomes available, updating the prediction made at $t = 10$ poses a challenge, as we only have the actual data for $t = 11$ while the data for $t = 12, 13, 14$ is still missing. This limitation hinders the model's ability to learn the most up-to-date information effectively, as seen in conventional time series forecasting.
>
> This challenge is further discussed in [*Section 4.2, line 409*], where DGrad, a simple method that eliminates information leakage by delaying ground truth feedback, does not yield optimal results and can lead to unreasonable large MSEs. Therefore, we proposed DSOF. We emphasize that DSOF is not a new model but a model-agnostic framework that enables deep time series models designed for batch learning to train online. Although teacher-student models and ER are standard approaches, from the results shown in [*Section 4.2, line 486*], it effectively reduces MSEs and maintain training stability.
>
>
> Here we provide an introduction of the mechanisms in DSOF for tackling the above challenge within the newly defined context:
> - DSOF addresses the challenges through a dual-stream approach. The fast stream processes the latest data immediately, without waiting for complete ground truth, and employs temporal difference (TD) learning to improve near-future forecasts. Meanwhile, the slow stream utilizes experience replay (ER) to stabilize the training process, reduce overfitting to noise, and ensure accurate generalization of the data distribution. This approach allows our method to effectively leverage the most recent data stream knowledge while maintaining robust generalization capabilities.

---

### Official Review · Reviewer_NzDP · 2024-11-03

**Soundness:** 3
**Presentation:** 3
**Contribution:** 3
**Rating:** 8
**Confidence:** 2

**Summary:**

The paper spots a problem in the training-evaluation pipeline of existing online time series forecasting studies. The problem is that  parts of the future sequence has already been used as labels for updating the model at previous time-steps. This results in information leakage in training. This information leakage results in an overly optimistic training loss and consequently a gap between training and causal evaluation (the latter is what we really care about). The main contribution of the paper is identifying this shortcoming of the previous works. The paper also presents a fast-slow dual network algorithm for OTSF in a setting with no information leakage. They also present comprehensive experiments to showcase the performance of their algorithm.

**Strengths:**

The identified information leakage problem is important, and I think would be influential for the field.

**Weaknesses:**

The fast learning part (Section 3.3.2) motivates by the necessity of fast updates to stay up-to-date with the recent data. For this purpose they update the student model using labels generated by the teacher. The teacher model is however out-dated. This seems like a contradiction, and it is not clear how it may resolve the problem of being up-to-date, except for the fact that the immediate next label, x_{t+1}, is used in updating the student. This seems very small updates

Alternatively, student updates could leverage k next real data for some k>1, at the cost of a delay of k time-steps. The student of the current work uses k=1. Alternatively, one could consider a whole spectrum (or hierarchy) of models with different values of k (for example powers of 2).

Computational complexity: the DSOF algorithm involves two networks. Moreover, the teacher model requires to generate a long sequence of samples at each time-step, which can be very computationally costly in certain architectures like decoder transformers. I wonder if the reported outperformance could be achieved under the same computational budget as the baselines?


Comments on the presentation:

The presentation could be much better. The text involves quite long sentences that reduce fluency, resulting in a tiring read. Fig. 1 does not help much in clarifying the information leakage problem. Despite of extensive use of indices, the arguments are vague in several places; for example in line 45 “the whole sequence” could be replaced by the indices you defined in the same sentence.
Appendices are missing.

The claimed new OTSF framework is not rigorously formalized. For example what data can or cannot be used at each time of the training to prevent information leakage.

The connection to RL and TD learning in section 3.3 seems very loose. It would be better not to use TD terminology.

The Appendices are missing from the pdf.

**Questions:**

Please see the comments under the Weakness section.

---

> ### Author Response · Authors · 2024-11-22
> **Response to Reviewer NzDP (Part 1)**
>
> ### Q1: Appendices
> > The Appendices are missing from the pdf.
>
> In the revised paper, the appendix has been incorporated at the end of the main text. Previously it was part of the supplementary materials.
>
> ### Q2: Connection to RL and TD learning
> > The connection to RL and TD learning in section 3.3 seems very loose. It would be better not to use TD terminology.
>
> We established a connection of TD learning in RL to DSOF in [*Section B, line 1205*]. We would be grateful if you could review it to see if it addresses your concerns.
>
> ### Q3:  Achieving DSOF performance with batch learning computational costs
> > Computational complexity: the DSOF algorithm involves two networks. Moreover, the teacher model requires to generate a long sequence of samples at each time-step, which can be very computationally costly in certain architectures like decoder transformers. I wonder if the reported outperformance could be achieved under the same computational budget as the baselines?
>
> There are batch learning methods in time series forecasting, such as RevIN [r1], focus on enhancing data generalization in testing domains without requiring online updates. While it may be possible to achieve good results on benchmark datasets without online updates, it is important to note that the objectives of batch learning methods and online learning methods differ.
>
> Batch learning methods aim to improve test-time generalization, making them suitable when we expect unseen data to come from a similar distribution as the training data. These methods excel at generalizing beyond the specific examples they were trained on. Conversely, online learning methods are more appropriate when test data may differ from the training distribution, as they can adapt to emerging correlations and patterns in the temporal domain that batch learning might miss.
>
> Hybrid approaches that combine elements of both paradigms are also feasible. For instance, models can be initially trained using generalization techniques and later fine-tuned or updated with online learning techniques as new data becomes available. This approach strikes a balance between strong initial generalization and adaptability to new information.
>
> ### Q4: About fast data stream
> > The fast learning part (Section 3.3.2) motivates by the necessity of fast updates to stay up-to-date with the recent data. For this purpose they update the student model using labels generated by the teacher. The teacher model is however out-dated. This seems like a contradiction, and it is not clear how it may resolve the problem of being up-to-date, except for the fact that the immediate next label, x_{t+1}, is used in updating the student. This seems very small updates.
>
> In our paper, we hypothesize that model accuracy improves when more recent data points are used. For instance, when the model is at $t = 10$, given data at $t = 6,7,8,9,10$, it predicts values for $t = 11, 12, 13, 14$. Once the ground truth for $t = 11$ becomes available, the teacher model can use data from $t = 7, 8, 9, 10, 11$ to predict $t = 12, 13, 14, 15$. The inclusion of the ground truth at $t = 11$ is expected to enhance the accuracy of predictions for $t = 12, 13, 14$ made at $t = 11$, compared to those made at $t = 10$. We combine the ground truth of $t = 11$ with the teacher's predictions for $t = 12, 13, 14$ to create pseudo labels for backpropagation.
>
> This approach is analogous to TD learning, where the value of the next state is used to update the current state, or to Deep Q-Networks (DQN) [r2], which update the policy network using the Q-value prediction of the next state from the target network. More information can be found in [*Section B, line 1205*]
>
> Additionally, we assumed that in the context of $t = 11$, predictions for $t = 13$ are likely to be less accurate than those for $t = 12$. To manage the increased uncertainty of predictions at later time steps, we introduce a regularization coefficient $\gamma$. This coefficient adjusts the contribution of the error to the training loss, helping to balance the impact of inaccuracies in predictions further into the future.
>
> ---
> [r1] T. Kim, J. Kim, Y. Tae, C. Park, J.-H. Choi, and J. Choo, “Reversible instance normalization for accurate time-series forecasting against distribution shift,” in International Conference on Learning Representations, 2022
>
> [r2] V. Mnih, “Playing atari with deep reinforcement learning,” arXiv preprint arXiv:1312.5602, 2013

---

> ### Author Response · Authors · 2024-11-22
> **Response to Reviewer NzDP (Part 2)**
>
> ### Q5: Introducing a parameter $k$ for constructing pseudo labels
> >Alternatively, student updates could leverage k next real data for some k>1, at the cost of a delay of k time-steps. The student of the current work uses k=1. Alternatively, one could consider a whole spectrum (or hierarchy) of models with different values of k (for example powers of 2).
>
> Thank you for your suggestion. The corresponding changes can be found in [*Section A.6.2, line 1100*] of the revised paper.
>
> Specifically, a new parameter $k$ is introduced in constructing a pseudo label. This parameter allows for adjusting the balance between ground truth data and the teacher model's predictions. The value $k$ indicates the use of the most recent $k$ ground truth data points and the first $H-k$ steps of teacher model's prediction to construct the pseudo label.
>
> In this experiment, instead of relying solely on the most recent data, we utilize a range of $k$ values, specifically powers of 2 that are less than the prediction sequence length as suggested.
>
> In the following table, the setting of using powers of 2 is denoted as $k = 2^p$, while the setting $k=1$ corresponds to the original method in the pseudo label construction. The results indicate that this method is particularly beneficial for larger values of $H$. For instance, applying to the ETTh2 dataset using PatchTST with prediction length $H = 96$ provides significant improvements.
>
> |             |    |  DLinear  |            |  FSNet  |            |  PatchTST |            |
> |-------------|:--:|:---------:|:----------:|:-------:|:----------:|:---------:|:----------:|
> | Datasets    |  H |  $k = 1$  | $k = 2^p$ | $k = 1$ | $k = 2 ^p $ |  $k = 1$  | $k = 2 ^p $ |
> | Electricity | 24 | **4.737** |    4.803   |  5.475  |  **5.323** | **5.169** |    5.191   |
> |             | 48 |   6.181   |  **6.022** |  7.000  |  **6.821** |   6.665   |  **6.566** |
> |             | 96 |   10.661  | **10.393** |  10.023 |  **9.839** |   11.248  | **10.249** |
> | ETTh2       | 24 | **1.701** |    1.713   |  3.114  |  **2.742** |   1.925   |  **1.874** |
> |             | 48 |   3.082   |  **3.015** |  5.318  |  **4.959** |   4.473   |  **3.970** |
> |             | 96 |   5.609   |  **5.323** |  9.526  |  **9.446** |   17.363  | **13.351** |

---

> > ### Comment · Reviewer_NzDP · 2024-11-28
> > **After reading the authors response:**
> >
> > Thanks for your response and the efforts in addressing the comments. In particular, the new set of experiments and discussion of the idea about utilizing a range of values of k for balancing between the ground-truth data and teacher’s predictions instead of relying solely on the most recent data, and the fact that these these experiments are conducted on longer range predictions than those reported in the main body of the paper. Also the discussions of information leakage idea in the introduction is improved. I wonder why Section 1.1 begins by a numeric example before discussing the general prediction setting, as in the previous version.
> >
> > There are however some important issues remaining, which are pretty straightforward to address. I would be happy to increase my score by 1-3 points if the authors could commit to further address the following issues.
> >
> > 1- I found the provided connection of DSOF to TD learning (in Appendix B) to be loose and erroneous. For example, value function $V(S)$ in RL is a scalar, but here you refer to a sequence of predictions (sequence of vectors) as the value function. Moreover, the present problem involves no dynamics in the sense that current action does not influence the next state, as opposed to RL. Please note that even though in DSOF the student somehow bootstraps from the teacher model, not every kind of bootstrapping is TD learning. I think the name teacher-student explains itself and is well established in the field, without a need to draw its connections to TD, etc. As mentioned in the previous round, I think the claim about connection to TD completely removed from the paper.
> >
> > 2- The improvements achieved through introducing the parameter k in Appendix A.6 are considerable. I wonder why the main body of the paper makes no mention or reference to this appendix? Could the authors add a brief discussion of Appendix A.6.2 (e.g., the summary of the first two paragraphs of A.6.2) to the main body of the paper?
> >
> > Further improvements that authors may consider to include in this paper or future works: it would be interesting to explore the optimal value of k.  More specifically, instead of summing over exponential values of k, use a single k and test for what value of k=1,…,H the best performance would be achieved. This will help to understand the optimal balance between ground-truth data and teacher’s predictions.

---

> ### Author Response · Authors · 2024-11-30
> **2nd Response to Reviewer NzDP (Part 1)**
>
> ### Q1: Connection of DSOF to TD Learning
>
> We acknowledge the fundamental differences between the settings of RL and OTSF. RL primarily focuses on the sequence of actions to maximize long-term rewards, whereas OTSF emphasizes short-term accuracy and continuous adaptation. Rather than completely transforming the OTSF problem into an RL framework, the term "TD" is used in this paper to indicate the inspiration from TD learning in RL, that it allows for intermediate parameter updates instead of waiting for the entire episode to finish. In the context of DSOF, "TD" highlights the process of making intermediate updates without waiting for all actual data to arrive, thus addressing the issue of missing ground truth in OTSF.
>
> Besides, Section B of the paper is not intended to offer a rigorous mathematical formulation. Instead, its aim is to highlight the similarities in the need for intermediate updates in both RL and OTSF. However, this intention is not clearly specified in the paper. Therefore, we propose to make the following adjustments to the paper:
>
> 1.  In the introduction section of the paper [*Section 1.2, line 109*], we can mention that TD learning serves as an inspiration (rather than being directly applied).
>
>     - This dual-stream mechanism benefits the online training process in adaptability and stability. The fast stream, inspired by TD learning's intermediate value updates, leverages the latest information from data streams to improve near-future forecasts without waiting for the complete ground truth sequence. Meanwhile, the slow stream conducts experience replay (ER) to stabilize training, penalizes overfitting the noise in the incoming data, and ultimately learn accurate generalization of the data distribution.
>
>
> 2. In the methodology section [*Section 3.3, line 230*], we clarify that the use of the "TD" terminology is intended to highlight the process of intermediate updates.
>
>     - This section presents the two data streams for updating the teacher-student model in Section 3.2: a slow stream that updates the model using complete ground truth data through ER and a fast stream that incorporates the latest information via a TD learning-like method in RL. The term "TD" indicates that our approach is inspired by TD learning in reinforcement learning, particularly in estimating the ground truth values at the next time step, addressing the challenge of missing ground truth in the redefined OTSF setting.
>
>
> 3. We can also update the introduction of [*Section B, line 1208*] to highlight the common challenges encountered by both TD learning and OTSF, and to explain how the principles of TD learning can be connected to OTSF.
>
>     - As outlined in Section 3.3.2, the fast stream leverages ideas from TD learning to quickly process latest data and improve near-future forecasts. While the learning goals of RL and OTSF differ, both fields share the challenge of updating models without complete access of ground truth information. In this section, drawing inspiration from TD learning in RL, we demonstrate how the idea of estimating the value of the next state can be applied to facilitate intermediate updates in OTSF.
>
> 4. At the conclusion of [*Section B, line 1236*], we can add a final remark to emphasize that this section does not provide an exact formulation of the OTSF problem as an RL problem, due to the inherently different objectives of the two fields, but rather seeks to enhance OTSF performance with techniques in TD learning.
>
>     - While insights from TD learning in RL can potentially enhance OTSF performance, OTSF should not be framed as an RL problem because their primary objectives are fundamentally different. RL focuses on optimizing sequence of actions for maximizing long-term rewards, whereas OTSF seeks accurate near-future predictions for data streams which may have changing distributions. This distinction leads to unique algorithmic settings; for instance, in RL, the current action influences the next state, whereas in OTSF, previous forecasts do not impact subsequent ones.
>
>
> ### Q2: Mentioning the introduction of $k$ in the main text
>
> We agree that this should be referenced in the main text. The introduction of parameter $k$ can be mentioned in [*Section 3.3.2, line 264*]:
>
> - In practice, the pseudo labels can be generated with different proportions of ground truth data and teacher model predictions. Consequently, we extend Equation 5 by introducing a parameter $k$, which signifies the use of the latest $k$ ground truth data points along with the initial $H-k$ steps of predictions from the teacher model to form the pseudo label. The impact of using different $k$ values, as well as utilizing multiple $k$ values, are discussed in Section A.6.2.

---

> ### Author Response · Authors · 2024-11-30
> **2nd Response to Reviewer NzDP (Part 2)**
>
> ### Q3: Exploring optimal values of $k$
> >  it would be interesting to explore the optimal value of k. More specifically, instead of summing over exponential values of k, use a single k and test for what value of k=1,…,H the best performance would be achieved. This will help to understand the optimal balance between ground-truth data and teacher's predictions.
>
> In [*Section A.6.2, line 1101*], we can insert a subsection to demonstrate the effects of altering the values of $k$:
>
> - In this experiment, we set $ H = 48$, with DLinear acting as the teacher model for DSOF. Our aim is to find the best ratio of ground truth values to the teacher model's predictions for constructing the pseudo label. We conduct the experiment by selecting a single value of $k$ instead of multiple values for the fast stream, and testing with evenly distributed $k$ values. Specifically, we chose multiples of 8, i.e. $k \in \{1, 8, 16, 24, 32, 40\}$. As shown in the table, smaller values of $ k $ are preferred. The results align with our expectations, indicating that the model should be updated more promptly with the most recent data points.
>
> | $k$ | ECL | ETTh2 |
> |---|:---:|:---:|
> | 1 | 6.181 | 3.082 |
> | 8 | **6.090** | 3.076 |
> | 16 | 6.209 | **3.042** |
> | 24 | 6.458 | 3.122 |
> | 32 | 6.798 | 3.332 |
> | 40 | 6.996 | 3.345 |
>
> ### Q4: Description of the setting in Section 1.1
> > I wonder why Section 1.1 begins by a numeric example before discussing the general prediction setting, as in the previous version.
>
> We were concerned that the notation was overly complex, impacting the clarity of the paper. However, we also agree that using such notations helps maintain the formality of the prediction setting. As a result, we are revising [*Section 1.1, line 041*] to integrate both the notations and the example for better understanding:
>
> - Existing works that endeavor to improve OTSF model accuracies (Pham et al., 2023; Zhang et al., 2023; 2024a) follow a uniform task setting: at time $t = i$, the model receives input data with timestamps from $t = i-L-H+2$ to $ t = i-H+1$ and generates predictions from $ t = i-H+2$ to $ t = i+1$; at the next timestamp $ t = i+1$, the actual data for $ t = i+1$ becomes available, allowing the model to evaluate its previous prediction and adjust its parameters based on this new ground truth. For instance, as illustrated in Figure 1, if the input and output windows are of lengths $L = 5$ and $H = 4$ respectively, at time $t=10$, the model first takes in data from $t = 3$ to $t= 7$ and ... (The rest of the content remains identical to the main text)

---

> ### Comment · Reviewer_NzDP · 2024-11-30
>
> Thanks to the authors for their response. My concerns about the optimal k and the first paragraphs of the introduction are addressed. The claimed connection to TD is still a pushback, because the justification in the appendix is loose and erroneous. I am inclined to raise my score to 7. However since the scores are quantized at 6 and 8, I raise the score to 8 and keep my confidence at 2.

---

### Author Response · Authors · 2024-11-22
**Summary of Paper Revision**

The following amendments have been made to the manuscript:

1. The appendix is now included at the end of the main text. Previously, it was separated into the supplementary materials, which is inconvenient due to broken hyperlinks.

1. Purple text indicates revisions made to the paper.

1. Description of information leakage problem in OTSF
    1. Using concrete examples instead of mathematical notation to describe the problem for easier understanding [*Section 1.1, line 041*]
    1. New figure for illustrating the information leak problem [*Section 1.1, line 062*]
    1. Moving the formal definition of the redefined setting from introduction to methodology [*Section 3.1, line 191*]
    1. Moving the formulation of the existing setting used in previous OTSF works from introduction to appendix [*Section C, line 1237*]

1. MSE Comparison of Batch learning and DSOF
    1. Updated Results of OneNet [*Section 4.1, line 378*]
    1. Reporting both mean and standard deviation of repeated experiments [*Section E.1, line 1289*]

1. MSE comparison of DSOF with other online learning frameworks
    1. Change of name: the online learning framework used by FSNet and OneNet is renamed from "Prev." to "DGrad".  [*Section 4.2, line 411*]
    2. Included other online learning frameworks, TFCL and DER++, as baselines [*Section 4.2, line 419*]
    1. Runtime (wall time and processor time) comparison of DSOF [*Section A.1.1, line 686*]
    1. GPU memory usage comparison [*Section A.1.2, line 701*]

1. Ablation studies for the main components in DSOF
    1. The baseline for the ablation study, i.e. the case where both components are not used, is modified. Previously the baseline was batch learning, but now is replaced with DGrad. [*Section A.2.1, line 725* and *Section A.2.2, line 751*]
    1. Included case studies for the interaction of ER and TD loss components [*Section A.2.3, line 772*]
    1. Included runtime (wall time and processor time) comparison [*Section A.2.4, line 794*]

1. Added experiments for testing different batch sizes used in ER procedure. [*Section A.5.2, line 969*]
1. Introducing a parameter $k$ for constructing pseudo labels [*Section A.6.2, line 1100*]
1. Experiments on different look-back window lengths [*Section A.8, line 1173*]


Pending revisions for the upcoming versions of the paper:
1. About the connection of TD in RL and DSOF:
    1.  In the introduction section of the paper [*Section 1.2, line 109*], mention that TD learning serves as an inspiration.
    1. In the methodology section [*Section 3.3, line 230*], clarify that the use of the "TD" terminology is intended to highlight the process of intermediate updates.
    1. Update the introduction of [*Section B, line 1208*] to highlight the common challenges encountered by both TD learning and OTSF, and to explain how the principles of TD learning can be connected to OTSF.
    1. At the conclusion of [*Section B, line 1236*], add a final remark to emphasize that this section does not provide an exact formulation of the OTSF problem as an RL problem, due to the inherently different objectives of the two fields, but rather seeks to enhance OTSF performance with techniques in TD learning.

2. Mention the introduction of parameter $k$ in pseudo label construction in [*Section 3.3.2, line 264*].

3. In [*Section A.6.2, line 1101*], insert a subsection to demonstrate the effects of altering values of $k$.

4. To maintain the formality of the prediction setting as described in [*Section 1.1, line 041*], integrate both mathematical notations and a practical example.

---

> ### Author Response · Authors · 2024-12-02
>
> As the discussion period nears its end, we would like to express our gratitude to the reviewers for their dedication and effort in evaluating our work. We are truly thankful for the insightful feedback and valuable comments provided. Your constructive and instructive suggestions have been instrumental in guiding our revisions, as detailed in the above post. These improvements have enhanced the clarity of our paper, particularly in articulating the problem of information leakage for a broader audience. The reviews have also encouraged us to ensure comprehensiveness in our methodology, comparisons and analysis. Your contributions have significantly strengthened the quality of our paper.

---

### Meta-Review · Area_Chair_Pix8 · 2024-12-21

**Metareview:**

This paper identifies a common bug for evaluating online time series forecasting in which future sequence data used to evaluate predictions has been used to update the model at previous time steps. The authors also introduce a teacher-student model framework to update a model with real-time data, thus preventing data leakage during evaluation while allowing models to update incoming data.

Overall, this paper is quite significant because it identifies a very prevalent bug affecting the evaluation in a subfield. This contribution is unambiguously significant and presented well in the paper. The primary concerns about this paper relate to the student-teacher framework. The paper did not present the method clearly, and it was not obvious how the proposed method addressed the other identified problem of information leakage.

The authors made significant revisions to improve the paper's readability. The results evaluate online time series forecasting without information leakage, which is an important contribution in and of itself. For this reason, as well as the significance listed above, I vote that this paper be accepted.

**Additional Comments On Reviewer Discussion:**

Many reviewers brought up issues related to clarity and the presentation of the material. One reviewer wasn't sure if updating the student model using labels from the teacher model would not lead to out-of-date labels. Another reviewer wasn't sure how the method was related to the information leakage problem. The authors made a significant effort in the revision to address these issues and improve clarity.

There were some minor technical concerns about the experiments, including the inclusion of baselines and whether the proposed method had an unfair computational advantage due to the 2-model (student-teacher) setup. The authors sufficiently addressed these concerns in the rebuttal.

During the reviewer/AC discussion, reviewers commented on how the manuscript had been improved and that most concerns had been resolved. Without any significant lingering concerns, coupled with the significance of the information leakage problem, I decided to vote for acceptance.

---

### Decision · Program_Chairs · 2025-01-22

Accept (Poster)